# ɪLQR-VAE : control-based learning of input-driven dynamics with applications to neural data

**Marine Schimel**
Department of Engineering
University of Cambridge
Cambridge, UK
mmcs3@cam.ac.uk

**Ta-Chu Kao**
Gatsby Computational Neuroscience Unit
University College London
London, UK
c.kao@ucl.ac.uk

**Kristopher T. Jensen**
Department of Engineering
University of Cambridge
Cambridge, UK
ktj21@cam.ac.uk

**Guillaume Hennequin**
Department of Engineering
University of Cambridge
Cambridge, UK
g.hennequin@eng.cam.ac.uk

## Abstract

Understanding how neural dynamics give rise to behaviour is one of the most fundamental questions in systems neuroscience. To achieve this, a common approach is to record neural populations in behaving animals, and model these data as emanating from a latent dynamical system whose state trajectories can then be related back to behavioural observations via some form of decoding. As recordings are typically performed in localized circuits that form only a part of the wider implicated network, it is important to simultaneously learn the local dynamics and infer any unobserved external input that might drive them. Here, we introduce iLQR-VAE, a control-based approach to variational inference in nonlinear dynamical systems, capable of learning both latent dynamics, initial conditions, and ongoing external inputs. As in recent deep learning approaches, our method is based on an input-driven sequential variational autoencoder (VAE). The main novelty lies in the use of the powerful iterative linear quadratic regulator algorithm (iLQR) in the recognition model. Optimization of the standard evidence lower-bound requires differentiating through iLQR solutions, which is made possible by recent advances in differentiable control. Importantly, the recognition model is naturally tied to the generative model, greatly reducing the number of free parameters and ensuring high-quality inference throughout the course of learning. Moreover, iLQR can be used to perform inference flexibly on heterogeneous trials of varying lengths. This allows for instance to evaluate the model on a single long trial after training on smaller chunks. We demonstrate the effectiveness of iLQR-VAE on a range of synthetic systems, with autonomous as well as input-driven dynamics. We further apply it to neural and behavioural recordings in non-human primates performing two different reaching tasks, and show that iLQR-VAE yields high-quality kinematic reconstructions from the neural data.

## 1 Introduction

The mammalian brain is a complex, high-dimensional system, containing billions of neurons whose coordinated dynamics ultimately drive behaviour. Identifying and interpreting these dynamics is the focus of a large body of neuroscience research, which is being facilitated by the advent of new experimental techniques that allow large-scale recordings of neural populations (Jun et al., 2017; Stosiek et al., 2003). A range of methods have been developed for learning dynamics from data (Buesing et al., 2012; Gao et al., 2016; Duncker et al., 2019; Archer et al., 2015; Hernandez et al., 2018; She and Wu, 2020; Kim et al., 2021; Nguyen et al., 2020). These methods all specify a

generative model in the form of a flexible latent dynamical system driven by process noise, coupled with an appropriate observation model.

Importantly, neural recordings are typically only made in a small selection of brain regions, leaving many areas unobserved which might provide relevant task-related input to the recorded one(s). Yet, the aforementioned methods perform Bayesian inference of state trajectories directly, and therefore do not support inference of external input (which they effectively treat as process noise and marginalize out). Indeed, simultaneous learning of latent dynamics *and* inference of unobserved control inputs is a challenging, generally degenerate problem that involves teasing apart momentary variations in the data that can be attributed to the system's internal transition function, and those that need to be explained by forcing inputs. This distinction can be achieved by introducing external control in the form of abrupt changes in the latent state transition function, and inferring these switching events (Ghahramani and Hinton, 2000; Linderman et al., 2017). More recently, Pandarinath et al. (2018) introduced LFADS, a sequential variational autoencoder (VAE) that performs inference at the level of external inputs as well as initial latent states. The inferred inputs were shown to be congruent with task-induced perturbations in various reaching tasks in primates (Pandarinath et al., 2018; Keshtkaran and Pandarinath, 2019). Further related work is discussed in Appendix A.

Here, we introduce iLQR-VAE, a new method for learning input-driven latent dynamics from data. As in LFADS, we use an input-driven sequential VAE to encode observations into a set of initial conditions and external inputs driving an RNN generator. However, while LFADS uses a separate, bidirectional RNN as the encoder, here we substitute the inference network with an *optimization-based recognition model* that relies on the powerful iterative linear quadratic regulator algorithm (iLQR, Li and Todorov, 2004). iLQR solves an optimization problem that finds a mode of the exact posterior over inputs for the current setting of generative parameters. This ensures that the encoder (mean) remains optimal for every update of the decoder, thus reducing the amortization gap (Cremer et al., 2018). Moreover, having the recognition model be implicitly defined by the generative model stabilizes training, prevents posterior collapse (thus circumventing the need for tricks such as KL warmup), and greatly reduces the number of (hyper-)parameters.

While iLQR-VAE could find applications in many fields as a general approach to learning stochastic nonlinear dynamical systems, here we focus on neuroscience case studies. We first demonstrate in a series of synthetic examples that iLQR-VAE can learn the dynamics of both autonomous and input-driven systems. Next, we show state-of-the art performance on monkey M1 population recordings during two types of reaching tasks (O'Doherty et al., 2018; Churchland et al., 2010). In particular, we show that hand kinematics can be accurately decoded from inferred latent state trajectories, and that the inferred inputs are consistent with recently proposed theories of motor preparation.

## 2 METHOD

iLQR-VAE models a set of temporal observations, such as behavioural and/or neural recordings, through a shared input-driven nonlinear latent dynamical system (Figure S1). The input encapsulates both process noise (as in traditional latent dynamics models), initial inputs that set the initial condition of the dynamics, and any meaningful task-related control input. In this section, we describe the architecture of the generative model, and the control-based variational inference strategy used for training the model and making predictions. A graphical summary of the model can be found in Appendix B.

### 2.1 GENERATIVE MODEL

We consider the following generative model:

$$\text{latent state} \quad \boldsymbol{z}_{t+1} = f_\theta(\boldsymbol{z}_t, \boldsymbol{u}_t, t) \tag{1}$$

$$\text{observations} \quad \boldsymbol{o}_t | \boldsymbol{z}_t \sim p_\theta(\boldsymbol{o}_t | \boldsymbol{z}_t) \tag{2}$$

where $\boldsymbol{u}_t \in \mathbb{R}^m$, $\boldsymbol{z}_t \in \mathbb{R}^n$ and $\boldsymbol{o}_t \in \mathbb{R}^{n_o}$ are the input, latent state and observations at time $t$, respectively. Here, observations may comprise either neural activity, behavioural variables, or both – the distinction will be made later where relevant. We use the notation $\theta$ to denote the set of all parameters of the generative model. We use $\boldsymbol{u}_0$ to set the initial condition $\boldsymbol{z}_1 = f_\theta(\boldsymbol{0}, \boldsymbol{u}_0, 0)$ of the

network[1]. This way, the latent state trajectory of the network $\boldsymbol{z}(\boldsymbol{u}) = \{\boldsymbol{z}_1, \ldots, \boldsymbol{z}_T\}$ is entirely determined by the input sequence $\boldsymbol{u} = \{\boldsymbol{u}_0, \ldots, \boldsymbol{u}_T\}$ and the state transition function $f_\theta(\cdot)$, according to Equation 1. For $f_\theta(\cdot)$, we use either standard linear or GRU-like RNN dynamics (see Appendix C for details). For the likelihoods, we use Gaussian or Poisson distributions with means given by linear or nonlinear readouts of the network state of the form $\bar{\boldsymbol{o}}_t = h(\boldsymbol{C}\boldsymbol{z}_t + \boldsymbol{b})$ (Appendix D).

We place a Gaussian prior over $\boldsymbol{u}_{t\leq 0}$. We then consider two alternative choices for the prior over $\boldsymbol{u}_{t>0}$. The first is a Gaussian prior

$$p_\theta(\boldsymbol{u}_{t>0}) = \mathcal{N}(0, \boldsymbol{S}^2) \tag{3}$$

with $\boldsymbol{S} = \text{diag}(s_1, \ldots, s_m)$. In many settings however, we expect inputs to enter the system in a sparse manner. To explicitly model this, we introduce a second prior over $\boldsymbol{u}$ in the form of a heavy-tailed distribution constructed hierarchically by assuming that the $i^{\text{th}}$ input at time $t > 0$ is

$$u_{it} = s_i \epsilon_{it} \sqrt{\nu/\alpha_t} \tag{4}$$

where $s_i > 0$ is a scale factor, $\epsilon_{it} \sim \mathcal{N}(0,1)$ is independent across $i$ and $t$, and $\alpha_t \sim \chi_\nu^2$ is a shared scale factor drawn from a chi-squared distribution with $\nu$ degrees of freedom. Thus, inputs are spatially and temporally independent a priori, such that any spatio-temporal structure in the observations will have to be explained by the coupled dynamics of the latent states. Moreover, the heavy-tailed nature of this prior allows for strong inputs when they are needed. Finally, the fact that the scale factor is shared across input dimensions means that inputs are either all weak or potentially all strong at the same time for all input channels, expressing the prior belief that inputs come as shared events.

This hierarchical construction induces a multivariate Student prior at each time step:

$$p_\theta(\boldsymbol{u}_t) = \frac{\Gamma\left[(\nu+m)/2\right]}{\Gamma\left[\nu/2\right](\nu\pi)^{m/2}|\boldsymbol{S}|}\left[1 + \frac{1}{\nu}\boldsymbol{u}_t^T\boldsymbol{S}^{-2}\boldsymbol{u}_t\right]^{-(\nu+m)/2} \tag{5}$$

where $\boldsymbol{S} = \text{diag}(s_1, \ldots, s_m)$. Note that both $\boldsymbol{S}$ and $\nu$ are parameters of the generative model, which we will learn.

## 2.2 ILQR-VAE: A NOVEL CONTROL-BASED VARIATIONAL INFERENCE STRATEGY

To train the model, we optimize $\theta$ to maximize the log-likelihood of observing a collection of independent observation sequences $\mathcal{O} = \{\boldsymbol{o}^{(1)}, \ldots, \boldsymbol{o}^{(K)}\}$, or "trials", given by:

$$\log p_\theta(\mathcal{O}) = \sum_{k=1}^{K}\log\int p_\theta(\boldsymbol{o}^{(k)}|\boldsymbol{z}(\boldsymbol{u}))p_\theta(\boldsymbol{u})\,d\boldsymbol{u}. \tag{6}$$

As the integral is in general intractable, we resort to a variational inference strategy by introducing a recognition model $q_\phi(\boldsymbol{u}|\boldsymbol{o}^{(k)})$ to approximate the posterior $p_\theta(\boldsymbol{u}|\boldsymbol{o}^{(k)})$. Following standard practice (Kingma and Welling, 2013; Rezende et al., 2014), we thus train the model by maximizing the evidence lower-bound (ELBO):

$$\mathcal{L}(\mathcal{O}, \theta, \phi) = \sum_k \mathbb{E}_{q_\phi(\boldsymbol{u}|\boldsymbol{o}^{(k)})}\left[\log p_\theta(\boldsymbol{o}^{(k)}|\boldsymbol{u}) + \log p_\theta(\boldsymbol{u}) - \log q_\phi(\boldsymbol{u}|\boldsymbol{o}^{(k)})\right] \tag{7}$$

$$= \sum_k \mathbb{E}_{q_\phi(\boldsymbol{u}|\boldsymbol{o}^{(k)})}\left[\sum_{t=1}^{T}\log p_\theta(\boldsymbol{o}_t^{(k)}|\boldsymbol{z}_t) + \log p_\theta(\boldsymbol{u}_t) - \log q_\phi(\boldsymbol{u}_t|\boldsymbol{o}^{(k)})\right] \tag{8}$$

$$\leq \log p_\theta(\mathcal{O}). \tag{9}$$

with respect to both $\theta$ and $\phi$.

---

[1]Note that when $m < n$, $\boldsymbol{u}_0$ can only reach an $m$-dimensional subspace of initial conditions, which could be limiting. We can circumvent this problem by spreading $\boldsymbol{u}_0$ over multiple surrogate time bins before the start of the trial, i.e. introduce $\{\boldsymbol{u}_{-n/m}, \ldots, \boldsymbol{u}_{-2}, \boldsymbol{u}_{-1}, \boldsymbol{u}_0\}$ together with an appropriate dependence of $f_\theta$ on $t \leq 0$ in Equation 1, such that each of these surrogate inputs target a different latent subspace with purely integrating ("sticking") linear dynamics before $t = 1$.

Here, the main novelty is the use of an optimization-based recognition model. We reason that maximizing the exact log posterior, i.e. computing

$$\boldsymbol{u}^{\star}(\boldsymbol{o}^{(k)}) = \underset{\boldsymbol{u}}{\operatorname{argmax}} \ \log p_\theta(\boldsymbol{u}|\boldsymbol{o}^{(k)}) \tag{10}$$

$$= \underset{\boldsymbol{u}}{\operatorname{argmax}} \left[ \sum_{t=1}^{T} \log p_\theta(\boldsymbol{o}_t^{(k)}|\boldsymbol{u}) + \log p_\theta(\boldsymbol{u}_t) \right] \tag{11}$$

subject to the generative dynamics of Equations 1 and 2, is a standard nonlinear control problem: $\log p_\theta(\boldsymbol{o}_t^{(k)}|\boldsymbol{u})$ acts as a running cost penalizing momentary deviations between desired outputs $\boldsymbol{o}_t$ and the actual outputs caused by a set of controls $\boldsymbol{u}$, and $\log p_\theta(\boldsymbol{u}_t)$ acts as an energetic cost on those controls. Importantly, there exists a general purpose, efficient algorithm to solve such nonlinear control problems: iLQR (Li and Todorov, 2004; Appendix E). We thus propose to use a black-box iLQR solver to parameterize the mean of the recognition density $q_\phi(\boldsymbol{u}|\boldsymbol{o})$ for any $\boldsymbol{o}$, and to model uncertainty separately using a multivariate Gaussian density common to all trials. Therefore, we parametrize the recognition model as follows:

$$q_\phi(\boldsymbol{u}|\boldsymbol{o}) = \mathcal{N}(\boldsymbol{u}; \boldsymbol{u}^{\star}(\boldsymbol{o}), \boldsymbol{\Sigma}_{\text{s}} \otimes \boldsymbol{\Sigma}_{\text{t}}) \tag{12}$$
$$\text{with } \boldsymbol{u}^{\star}(\boldsymbol{o}) = \text{iLQRsolve}(\boldsymbol{o}, \theta). \tag{13}$$

where we use a separable posterior covariance (the Kronecker product of a spatial factor $\boldsymbol{\Sigma}_{\text{s}}$ and a temporal factor $\boldsymbol{\Sigma}_{\text{t}}$).

To optimize the ELBO, we estimate the expectation in Equation 8 by drawing samples from $q_\phi(\boldsymbol{u}|\boldsymbol{o}^{(k)})$ and using the reparameterization trick (Kingma et al., 2015) to obtain gradients. A major complication that would normally preclude the use of optimization-based recognition models is the need to differentiate through the mean of the posterior. In this case, this involves differentiating through an entire optimization process. Using automatic differentiation within the iLQR solver is in general impractically expensive memory-wise. However, recent advances in differentiable model predictive control enable implicit differentiation through iLQRsolve with a memory cost that does not depend on the number of iterations (Amos et al., 2018; Blondel et al., 2021; Appendix F).

## 2.3 COMPLEXITY AND IMPLEMENTATION

We optimize the ELBO using Adam (Kingma and Ba, 2014) with a decaying learning rate $\propto 1/\sqrt{i}$ where $i$ is the iteration number. Averaging over data samples can be easily parallelized; we do this here using the MPI library and a local CPU cluster. In each iteration and for each data sample, obtaining the approximate posterior mean through iLQR is the main computational bottleneck, with a complexity of $\mathcal{O}(T(n^3 + n^2 n_o))$. To help mitigate this cost, we find it useful to re-use the previously inferred control inputs to initialize each iLQRsolve.

## 3 EXPERIMENTS AND RESULTS

### 3.1 ILQR-VAE ENABLES FAST LEARNING OF DYNAMICS

Before demonstrating the method on a number of synthetic and real datasets involving ongoing external inputs, we begin with a simpler example meant to illustrate some of iLQR-VAE's main properties (Figure 1). We generated data from an autonomous (i.e. non-input-driven) linear dynamical system ($n = 8$ latents, $m = 3$ input channels) seeded with a random initial condition in each of 56 trials. The state $\boldsymbol{z}_t$ was linearly decoded with added Gaussian noise to produce observation data, which we used to train a model in the same class.

At the beginning of learning, iLQR-VAE originally relies on large ongoing inputs that control the generator into producing outputs very similar to the observations in the data (Figure 1, red box, left), resulting in a rapidly decreasing loss. Subsequently, the amount of input required to fit the observations gradually decreases as the system learns the internal dynamics of the ground truth system. Eventually, the inferred control inputs become confined to the first time bin, i.e. they act as initial conditions for the now autonomous dynamics of the generative model. Thus, iLQR-VAE operates in a regime where the output of the generator explains the data well at all times, and learning

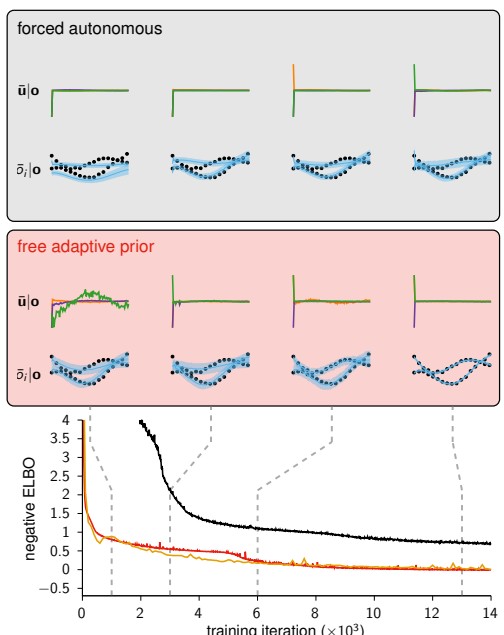

Figure 1: **Fast and robust learning in iLQR-VAE.** Example run of iLQR-VAE on a synthetic dataset generated by an autonomous linear dynamical. iLQR-VAE with an adaptive prior over control inputs (red) initially uses large inputs to fit the observations, but gradually pushes those inputs back into initial conditions as it acquires the ground truth autonomous dynamics. In contrast, iLQR-VAE with a rigid input prior imposing autonomous dynamics gets stuck in plateaus and learns considerably more slowly (black; see text for details). For each setting, insets show the three inferred inputs for a given test trial (top; posterior mean $\bar{u}|o$, rescaled by the maximum input in the sequence), and the posterior predictions for the first two corresponding outputs ($\bar{o}_i|o$; black dots: ground truth; blue: posterior mean with 95% c.i.). We also compare learning curves with LFADS (yellow curve). We used the same generator architecture in all scenarios, and the learning rate was hand-tuned for each example.

consists in making the inputs more parsimonious. We note that this regime is facilitated here by our choice of generator dynamics, which we initialised to be very weak (i.e with a small spectral radius) initially and therefore easily controllable.

We contrast this with learning in a modified version of iLQR-VAE where we allowed $u_0$ to vary freely (with a Gaussian prior of adjustable variance) but effectively fixed $u_{t>0}$ to be 0. In other words, we constrained the dynamics of the generator to remain (near-)autonomous throughout learning (Figure 1, grey box, top). Although this incorporates important information about the ground truth generator (which is itself autonomous), counter-intuitively we found that it impairs learning. At the beginning of training, iLQR is unable to find initial conditions that would explain the data well, resulting in a much higher initial loss. The model then gets stuck in plateaus that are seemingly avoided by the free version of iLQR-VAE (see Figure S2 for independent repeats of this experiment).

On the same toy dataset, we also compared iLQR-VAE to LFADS (Pandarinath et al., 2018), keeping the generative model in the same model class (see Appendix G for details). We found that LFADS learns in a similar manner to iLQR-VAE (Figure 1), also progressively doing away with inputs.

### 3.2 iLQR-VAE FOR NONLINEAR SYSTEM IDENTIFICATION

Next, we illustrate the method on an autonomous nonlinear dynamical system, the chaotic Lorenz attractor (Lorenz, 1963; Appendix H). This is a standard benchmark to evaluate system identification methods on nonlinear dynamics (Nguyen et al., 2020; Hernandez et al., 2018; Champion et al., 2019), and one typically considers the dynamics to be learned if the trained model can recreate the whole attractor structure.

Here, we show that iLQR-VAE can learn these complex nonlinear dynamics. Before training, the inferred inputs are large throughout the trial, and explain the output observations by forcing the internal state of the generator into appropriate trajectories (Figure 2A, top). At the end of learning, the inputs remain confined to the first time bin, setting the initial condition of the trajectories which are now driven by the stronger, near-autonomous dynamics of the generator. In Figure 2B we show that, conditioned on an initial bout of test data, the model perfectly predicts the rest of the trajectory. Moreover, starting from a random initial condition, the model can recreate the whole attractor structure (Figure 2C).

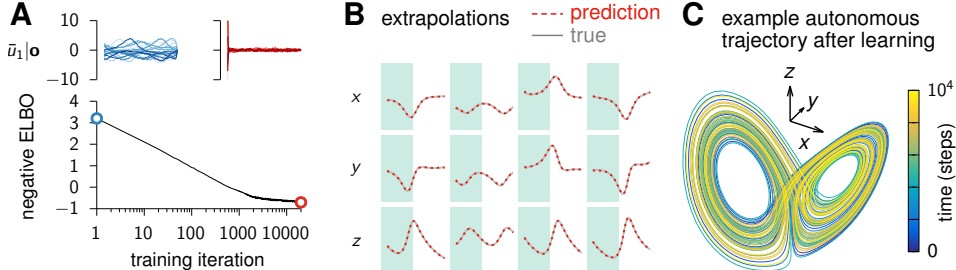

Figure 2: **Learning nonlinear autonomous dynamics. (A)** Evolution of the negative ELBO during training of an iLQR-VAE model with a nonlinear RNN and a Gaussian likelihood ($n = 20$, $m = 5$, $n_o = 3$). The inputs inferred by iLQR are originally strong (blue, before learning), but are progressively pushed to initial conditions (red, after learning) as the autonomous dynamics of the Lorenz attractor are approximated with increasing accuracy. **(B)** Four example test trajectories (columns), conditioning on the noisy data within the first half (green shading), and predicting in the second half. **(C)** Single long autonomous trajectory after training (setting $\boldsymbol{u}_{t>0} = 0$), starting from a random initial condition and running the dynamics for 10000 steps. The model displays the butterfly topology characteristic of the Lorenz system, and completes multiple cycles without deviating from the attractor, suggesting that the ground-truth dynamics have been learned.

To quantitatively assess how well the dynamics have been learned, we computed the $k$-step coefficient of determination, $R_k^2$, as in Hernandez et al. (2018). This metric evaluates how well the model can predict the true state $k$ steps into the future, starting from any state inferred along a test trajectory (see Appendix H for details). Hernandez et al. reported $R_{30}^2 \approx 1$ but did not show results for larger $k$. For iLQR-VAE, $R_{30}^2 = 0.998$ and the forward interpolation was still very high at 50 time steps, with $R_{50}^2 = 0.996$.

### 3.3 INFERRING SPARSE INPUTS

To demonstrate iLQR-VAE's ability to infer unobserved inputs and learn the ground truth dynamics of an input-driven system, we generated synthetic data from a system with $n = 3, m = 3$ and $n_o = 10$, which evolves with linear dynamics for $T = 1000$ time steps (see Appendix I for an example with input-driven *nonlinear* dynamics). The system was driven by sparse inputs, and the output corrupted with Gaussian noise. Input events were drawn in each time step from a Bernoulli distribution with mean $p = 0.03$. Whenever an input event occurred, the magnitude of inputs in each channel was drawn from a standard mutivariate Gaussian distribution.

We fit both iLQR-VAE and LFADS models to these data, choosing the generator to be within the ground-truth model class for both. iLQR-VAE captured most of the variance in the inputs (Figure 3A; $R^2 = 0.94 \pm 0.02$; 5 random seeds), and recovered the eigenvalue spectrum of the transition matrix almost perfectly (Figure 3B). LFADS however performed poorly on this example ($R^2 = 0.05 \pm 0.02$ for input reconstruction; 3 random seeds), as well as in several other similar comparisons on datasets of different sizes and trial numbers (Appendix J). This is unsurprising, as LFADS assumes a dense (auto-regressive) Gaussian prior over the inputs, which is not overridden by the relatively small amount of data used here. Note however that when applied to a set of 56 trials of 100 time steps driven by Gaussian autoregressive inputs, iLQR-VAE still captured the structure in the inputs more accurately than LFADS did ($R^2 = 0.81 \pm 0.01$ vs. $0.29 \pm 0.06$). We hypothesize that this reflects the difficulty of learning a good recognition model from a small amount of data. We evaluate the effect of the choice of prior more extensively in Appendix K.

### 3.4 PREDICTING HAND KINEMATICS FROM PRIMATE M1 RECORDINGS

#### 3.4.1 TRIAL-STRUCTURED MAZE TASK

To highlight the utility of iLQR-VAE for analyzing experimental neuroscience data, we next applied it to recordings of monkey motor (M1) and dorsal premotor (PMd) cortices during a delayed reaching task ('Maze' dataset of Kaufman et al., 2016; DANDI 000128). This dataset contains 108 differ-

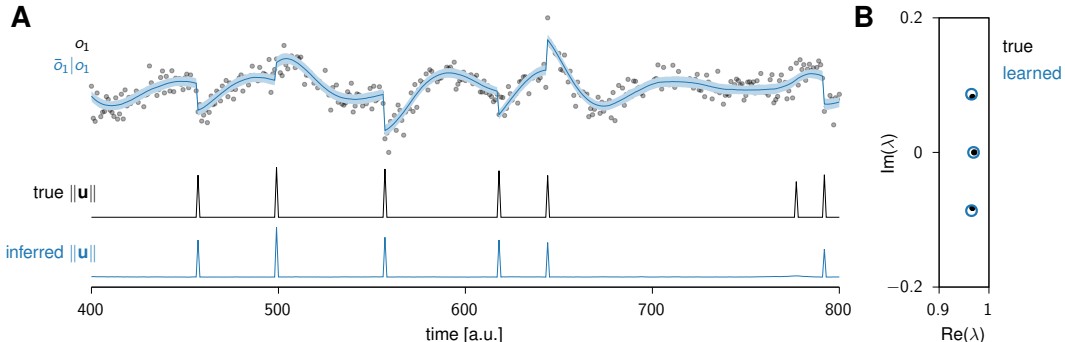

Figure 3: **Inferring sparse inputs to a linear system.** **(A)** Top: example observations (black dots) and inferred posterior mean (blue line). Bottom: true and inferred inputs. iLQR-VAE is able to infer the timing and magnitude of the inputs almost perfectly, despite being trained on only a single timeseries of 1000 time steps. Note that iLQR-VAE fails to infer one of the smallest inputs, whose effect on the observations is largely masked by observation noise. **(B)** Comparison of the true (black) and learned (blue) eigenvalue spectra. This shows that iLQR-VAE recovers the ground-truth dynamical system up to a similarity transformation.

ent reach configurations over nearly 3000 trials, and has recently been proposed as a neuroscience benchmark for neural data analysis methods (Pei et al., 2021). We compared the performance of iLQR-VAE to several other latent variable models, evaluated on this dataset in Pei et al. (2021).

Consistent with previous findings (Pandarinath et al., 2018), iLQR-VAE inferred inputs that were confined to initial conditions, from which smooth single-trial dynamics evolved near-autonomously (Figure 4A). As a first measure of performance, we evaluated the models on "co-smoothing", i.e the ability to predict the activity of held-out neurons conditioned on a set of held-in neurons (see Appendix L for details). Conditioning of 137 neurons (i.e using 45 held-out neurons), we obtained a co-smoothing of $0.331 \pm 0.001$ (over 5 random seeds). For comparison, Pei et al. (2021) reports 0.187 for GPFA (Yu et al., 2009), 0.225 for SLDS (Linderman et al., 2017), 0.329 for Neural Data Transformers (Ye and Pandarinath, 2021) and $R^2 = 0.346$ for AutoLFADS (LFADS with large scale hyperparameter optimization; Keshtkaran et al., 2021) on the same dataset.

Next, we assessed how well hand velocity could be decoded from neural activity – another metric of interest to neuroscientists. We applied ridge regression to predict the monkey's hand velocity (with a 100 ms lag) from momentary neuronal firing rates (mean of the posterior predictive distribution) on test data. This reconstruction could be performed with very high accuracy $R^2 = 0.896 \pm 0.002$ (over 5 random seeds), compared to 0.640 for GPFA, 0.775 for SLDS, 0.897 for Neural Data Transformers and 0.907 for AutoLFADS (Pei et al., 2021). These experiments place iLQR-VAE on par with state-of-the-art methods, without any extensive hyperparameter optimization.

### 3.4.2 CONTINUOUS REACHING TASK

While a large number of neuroscience studies perform neural and behavioural recordings during trial-structured tasks, much can be learned by analyzing the dynamics of more naturalistic, less constrained behaviours. iLQR-VAE's flexible recognition model is well-suited to the analysis of such less structured tasks, as it can easily be trained and tested on trials of heterogeneous lengths. To illustrate this, we applied iLQR-VAE to a self-paced reaching task during which a monkey had to reach to consecutive targets randomly sampled from a 17x8 grid on a screen (O'Doherty et al., 2018; Makin et al., 2018). This dataset consists of both neural recordings from primary motor cortex (M1) together with continuous behavioural recordings in the form of $x$- and $y$-velocities of the fingertip.

In this example, we experimented with fitting the spike trains and hand velocities jointly (combining a Poisson likelihood for the 130 neurons and a Gaussian likelihood for the 2 kinematic variables, see Appendix M for further details). We found that it allowed iLQR-VAE to reach a similar kinematic decoding performance as when fitting neural activity alone, but using a smaller network. More generally, we reason that a natural approach to making behavioural predictions from neural data using a probabilistic generator is to fit it to both jointly, and then use the posterior predictive distribution

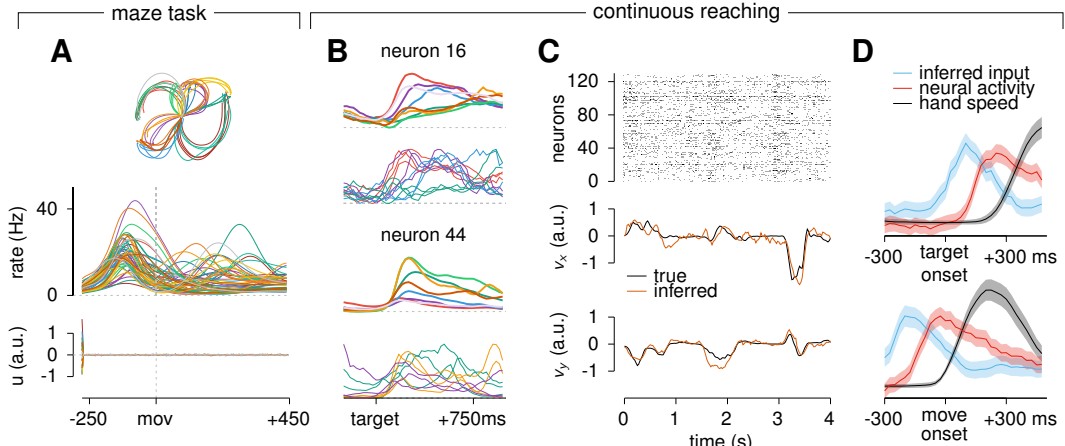

Figure 4: **iLQR-VAE can be used to decode kinematics from neural data, and learns dynamics relying strongly on preparatory inputs.** (**A**) 50 example hand trajectories (top) from the monkey reaching 'Maze' dataset, corresponding single-trial firing rate timecourse (middle; one example neuron), and inferred input (bottom; one example input dimension). (**B**) Mean (thick) and single-'trial' (thin) firing rates of two example neurons during reaches of various directions (colours), aligned to target onset. Note that three single-'trial' firing rates are shown for only three of the 8 reach directions for which averages are shown. Interestingly, single 'trial' activities evolve tightly around their trial averages, and resembles the firing rate responses shown in A (middle). (**C**) Example spike raster (top) and hand kinematics (bottom) for a 4 second-long chunk of test data in the continuous monkey reaching task. $v_x$ and $v_y$ refer to hand x- and y-velocities respectively. (**D**) Overall magnitude of the inferred inputs $\|\boldsymbol{u}_t\|$ (blue), average population spiking activity (red), and hand velocity $\|\boldsymbol{y}_t\|$ (black), each z-scored, averaged across movement episodes, and aligned to target onset (top) or movement onset (bottom).

over behavioural variables (conditioning on spike trains only) as a nonlinear decoder. In future work, this could provide more accurate predictions in those motor tasks where linear regression struggles (see e.g. Schroeder et al., 2022).

For our analyses, we used the first $\sim 22$ minutes of a single recording session (indy_20160426), excluded neurons with overall firing rates below 2 Hz, and binned data at 25 ms resolution. Although it is not a formal requirement of our method, we chunked the data into 336 non-overlapping pseudo-trials of 4 s each, in order to enable parallelization of the ELBO computation during training. We only trained the model on a random subset of 168 trials.

To highlight the flexibility of iLQR as a recognition model, we then evaluated the model by performing inference on the first 9 minutes of the data, as a single long chunk of observations. Note that this is not generally possible in LFADS or any sequential VAE where an encoder RNN has been trained exclusively on trials of the same fixed length. Despite the lack of trial structure, we found that neurons display a stereotyped firing pattern across multiple instances of each reach. This was revealed by binning the angular space into 8 reach directions, temporally segmenting and grouping the inferred firing rates according to the momentary reach direction, and aligning these segments to the time of target onset (Figure 4B). Moreover, hand kinematics could be linearly decoded from the inferred firing rates with high accuracy (Figure 4C; $R^2 = 0.75 \pm 0.01$ over 5 random seeds), on-par with AutoLFADS ($R^2 = 0.76$; Keshtkaran et al., 2021), and considerably higher than GPFA and related approaches ($R^2 = 0.6$; Jensen et al., 2021).

We next wondered whether we could use iLQR-VAE to address an open question in motor neuroscience, namely the extent to which the peri-movement dynamics of the motor cortex rely on external inputs (possibly from other brain areas). Such inputs could arise during movement preparation, execution, neither, or both. We thus examined the relationship between the inputs inferred by iLQR-VAE and the concurrent evolution of the neuronal firing rates and hand kinematics. Overall, neuronal activity tends to rise rapidly starting 150 ms before movement onset (Figure 4D, red), consistent with the literature (Shenoy et al., 2013; Churchland et al., 2012). Interestingly, we observed

that inputs tend to arise much earlier (around the time of target onset), and start decaying well before the mean neural activity has finished rising (Figure 4D top), about 150 ms before the hand started to move (Figure 4D, bottom). While these results must be interpreted cautiously, as inference was performed using information from the whole duration of the trial (i.e. using iLQR as a smoother), they show that the data is best explained by large inputs prior to movement onset, rather than during movement itself. Interestingly, the timing of these inputs is globally consistent with target-induced visual inputs driving preparatory activity in M1, whose dynamics then evolve in a more autonomous manner to drive subsequent motion.

## 4 DISCUSSION

### LIMITATIONS AND FUTURE WORK

While we have demonstrated that iLQR-VAE performs well on various toy and real datasets, the method has a number of limitations, some of which could be addressed in future work. Firstly, the problem of decoupling ongoing inputs from dynamics is degenerate in general, and there is no guarantee that iLQR-VAE will always successfully identify the ground-truth. This problem will be exacerbated in the low data regime, or if there is a large mismatch between our prior over inputs and the true input distribution. While further generalization tests such as extrapolations can be used to assess post-hoc how well the dynamics have been learned, the lack of identifiability will often make interpretation of the model parameters difficult. Secondly, using iLQR as a way of solving maximum a posteriori inference in state space models comes at a high computational cost, and with the risk that iLQR may converge to a local minimum. We note that both these issues could potentially be tackled at once if process noise in the generator was modelled separately from control inputs, as the MAP estimation problem could then be solved using some of the highly efficient algorithms available in the framework of linearly solvable stochastic control (Todorov, 2009; Dvijotham and Todorov, 2013; Kappen, 2005). Finally, for simplicity we modelled posterior input uncertainty using a common covariance across all data samples. This might be limiting, for example when modelling neural populations that exhibit coordinated global firing fluctuations giving rise to data samples with highly variable information content. A better solution, left to future work, would be to amortize the computation of the posterior uncertainty by reusing some of the computations performed in iLQR.

### CONCLUSION

The rise of new tools and software now makes it possible to record from thousands of neurons while monitoring behaviour in great detail (Jun et al., 2017; Mathis et al., 2018; Musk et al., 2019). These datasets create unique opportunities for understanding the brain dynamics that underlie neural and behavioural observations. However, identifying complex dynamical systems is a hard nonlinear filtering and learning problem that calls for new computational techniques (Kutschireiter et al., 2020). Here, we exploited the duality between control and inference (Toussaint, 2009; Kappen and Ruiz, 2016; Levine, 2018; Appendix N) to bring efficient algorithms for nonlinear control to bear on learning and inference in nonlinear state space models.

The method we proposed uses iLQR, a powerful general purpose nonlinear controller, to perform inference over inputs in an RNN-based generative model. Using an optimization-based recognition model such as iLQR has two advantages. First, it brings important flexibility at test time, enabling predictions on arbitrary, heterogeneous sequences of observations as well as seamless handling of missing data. Second, owing to parameter sharing between the generative and recognition models, the ELBO gap is reduced (Appendix O), making learning more robust (in particular, to initialization) and reducing the number of hyperparameters to tune. With the advent of automatically differentiable optimizers (Blondel et al., 2021), we therefore hope that optimization-based recognition models will open up new avenues for VAEs.

### ACKNOWLEDGEMENTS

We thank Jasmine Stone and Javier Antorán for helpful comments on the manuscript, and Anil Madhavapeddy for providing computing resources at early stages. M.S. was funded by an EPSRC DTP studentship and K.T.J. was funded by a Gates Cambridge scholarship. This work was performed

using resources provided by the Cambridge Tier-2 system operated by the University of Cambridge Research Computing Service (http://www.hpc.cam.ac.uk) funded by EPSRC Tier-2 capital grant EP/P020259/1.

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

# Appendix

## A    ADDITIONAL RELATED WORK

In this section, we first discuss (non-exhaustively) several methods used for identifying dynamical systems from data, before presenting the few approaches we are aware of that explictly tackle the problem of inferring unobserved control inputs to those systems.

The problem of identifying the dynamics giving rise to a set of observations is one that spans many fields, from climate modelling to neuroscience, and a variety of methods have therefore been developed to tackle it. Most existing approaches assume non-driven dynamics, as this greatly facilitates systems identification.

One common modelling paradigm is to assume the data arises from a latent linear dynamical system (LDS), which parameters can be learned using an Expectation-Maximization (EM) approach Ghahramani and Hinton (1996). While linear models are typically very efficient as they allow estimates to be computed in closed-form, they severely restrict the range of dynamics that can be approximated. Various extensions have been proposed, such as switching linear dynamical systems (Linderman et al., 2017; Ghahramani and Hinton, 2000), which assume that the data can be modelled using several latent dynamical systems with a Hidden Markov Model controlling the transitions between those. Alternatively, Costa et al. (2019) proposes to use adaptative locally linear dynamics and uses an iterative procedure to find the most likely switching points. In a similar vein, Hernandez et al. (2018) approximates the dynamics as locally linear; interestingly, the proposed method (VIND) incorporates the generative dynamics in the approximate posterior distribution over latent trajectories given data. This is reminiscent of the approach taken in iLQR-VAE, where the recognition parameters are kept tied to the generative parameters.

Another way to keep the problem solvable while allowing for richer dynamics is to approximate those using a linear combination of nonlinear basis functions. This then turns the optimization into the more simple problem of learning the weights of the expansion (with the caveat that one needs to choose the set of basis functions). This is the method used in Brunton et al. (2016b), with an additional constraint that the coefficients are sparse in the space of basis functions, yielding a more interpretable model. This was later extended in Champion et al. (2019) to allow for automatic discovery of a set coordinates in which the dynamics can be approximated as sparse.

In a similar manner, a popular approach involves modelling the dynamics as linear in the space of observables (which can include linear or nonlinear mappings from the state of the system), as is done in dynamic mode decomposition Schmid (2010); Kutz et al. (2016) (see Brunton et al. (2016a) for applications to neural data). This approach is closely related to Koopman operatory theory, which finds a set of dynamic modes and uses those to approximate the data as a single linear dynamical system.

Finally, the dynamics can be modelled using nonlinear neural networks, and the parameters learned using variational methods (see e.g Nguyen et al., 2020; Hernandez et al., 2018; Koppe et al., 2019).

Most of the aforementioned models can be extended to incorporate known external inputs coming into the system. This is for instance done in dynamic mode decomposition with control inputs (DMDc; Proctor et al., 2016), which can be generalized into Koopman operators with inputs and control (KIC; Proctor et al., 2018).

On the other hand, the range of methods modelling dynamics driven by *unobserved inputs* (which must thus be inferred) is a lot more limited. Indeed, LFADS (Pandarinath et al., 2018) is the first method we are aware of which explicitly models the set of control inputs driving the system. As described in the main text, LFADS models the dynamics as a (potentially) input-driven nonlinear dynamical system, and learns both the parameters and the inputs. More recently, Fieseler et al. (2020) proposed an extension of DMDc to handle unsupervised learning of unobserved signals as well as estimation of the dynamics. This was then used this to successfully model neural recordings made in C. elegans. Crucially however, the dynamics were modelled as linear, thus restricting the range of dynamics that the learnt system could generate. Morrison et al. (2020) modelled the same data using input-driven nonlinear dynamics, but assumed a limited subset of inputs driving

transitions at given time points, and thus only learned the magnitude of those inputs and not their timing.

Finally, an approach related to the modelling of unobserved inputs (which give rise to changes that cannot be explained by the dynamics alone) is the explicit modelling of events which lead to discontinuities in the dynamics. This is done in Chen et al. (2020) within the framework of neural ordinary differential equations (Chen et al., 2018). To some extent, one can also view switching dynamical models as inferring unobserved inputs giving rise to state transitions, although those "inputs" are restricted to live in a discrete subspace.

## B  GRAPHICAL SUMMARY OF THE MODEL

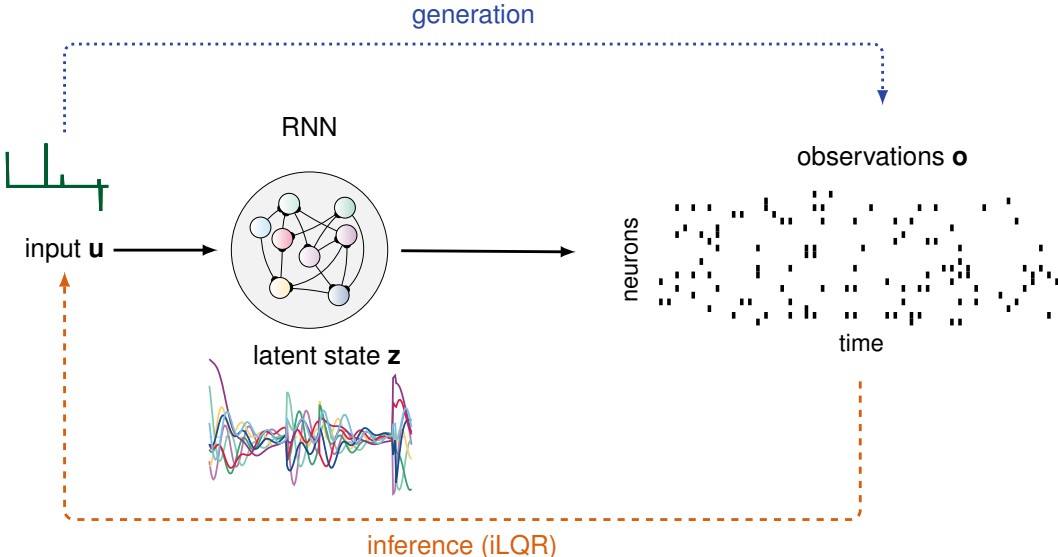

Figure S1: **Illustration of the model** iLQR-VAE is trained to model a set of noisy observations (here, spike trains). During each training iteration, iLQR is used to infer the input $u$ (green), given observations and the current parameters of the generator. Intuitively, the optimal inputs are ones that produce latent trajectories in the RNN that are most compatible with the data, without overfitting. Predictions are generated by running the dynamics forward, conditioned on a given input and set of parameters.

## C  IMPLEMENTATION OF THE DYNAMICS

We considered different functional forms for the discrete-time dynamics of the latent state. In the following, $z_t$ and $u_t$ denote the latent state and an external input at time t, respectively.

### C.1  LINEAR DYNAMICS

The simplest case considered is that of linear dynamics:

$$z_{t+1} = Az_t + Bu_t \tag{S1}$$

One issue with linear dynamics is that they may become unstable, such that repeated application of the operator $A$ will lead to a divergence of $\|z\|$ and the associated gradients. This can become problematic, especially when modelling long sequences of observations. To circumvent this issue, we used a parametrization of the propagator $A$ that ensured it remained stable at all times. To find a stable linear parametrization, we considered the Lyapunov stability condition (Bhatia and Szegö, 2002). The discrete time dynamics of Equation S1 are asymptotically stable if and only if $A$ satisfies

$$P - APA^T = I \tag{S2}$$

for some positive definite matrix $\boldsymbol{P}$ with eigenvalues $\geq 1$. It is easy to verify that the following parameterization of the state matrix $\boldsymbol{A}$ satisfies this criterion:

$$\boldsymbol{A} = \boldsymbol{U}\boldsymbol{D}^{1/2}\boldsymbol{Q}(\boldsymbol{D} + \boldsymbol{I})^{-1/2}\boldsymbol{U}^T \tag{S3}$$

with $\boldsymbol{U}$ and $\boldsymbol{Q}$ arbitrary unitary matrices, and $\boldsymbol{D}$ an arbitrary non-negative diagonal matrix. Conversely, any stable matrix can be reached by this parameterization. Note that the matrix $\boldsymbol{P}$ that satisfies Equation S2 is then given by $\boldsymbol{P} = \boldsymbol{U}(\boldsymbol{D} + \boldsymbol{I})\boldsymbol{U}^T$. Finally, as we are also learning the $\boldsymbol{B}$ and $\boldsymbol{C}$ matrices in Equation S1, we can without loss of generality set $\boldsymbol{U} = \boldsymbol{I}$.

## C.2 GRU DYNAMICS

To fit the monkey reaching data as well as the Lorenz attractor, we chose the dynamical system to be Minimal Gated Unit (MGU). More specifically, we used the MGU2 variant of the MGU proposed in Heck and Salem (2017):

$$\boldsymbol{f}_t = \sigma(\boldsymbol{U}_f \boldsymbol{z}_{t-1}) \tag{S4}$$

$$\hat{\boldsymbol{z}}_t = g(\boldsymbol{U}_h(\boldsymbol{f}_t \odot \boldsymbol{z}_{t-1}) + \boldsymbol{W}\boldsymbol{x}_t + \boldsymbol{b}_h) \tag{S5}$$

$$\boldsymbol{z}_t = (1 - \boldsymbol{f}_t) \odot \boldsymbol{z}_{t-1} + \boldsymbol{f}_t \odot \hat{\boldsymbol{z}}_t \tag{S6}$$

where $\boldsymbol{x}_t = \boldsymbol{B}\boldsymbol{u}_t$ denotes the input entering the dynamical system. Note that the latent state $\boldsymbol{z}$ is often denoted by $\boldsymbol{h}$ is the literature. We found that the MGU2 gave better and more stable performance than the MGU. We hypothesize that this is due to the input entering the system in the update gate only (as opposed to entering it through both forget and update gates), thus making the system more easily controllable. We chose $\sigma(\cdot)$ to be a sigmoid function, and $g(\cdot)$ to be a soft ReLu-like nonlinearity,

$$g(x) = \frac{x + \sqrt{x^2 + 4}}{2} - 1. \tag{S7}$$

# D LIKELIHOOD FUNCTIONS

The likelihood of the observations appears both in the ELBO and in the iLQR cost. Minimization of the latter via iLQR requires computing the momentary Jacobians and Hessians of the likelihood function w.r.t. the internal state $\boldsymbol{z}_t$. Although these quantities can be obtained generically via automatic differentiation, iLQR is always faster when they are provided directly (Appendix E), which we did here using the analytical expressions given below.

## D.1 GAUSSIAN LIKELIHOOD

For the Gaussian likelihood, we assume observations $\boldsymbol{o}$ are linearly decoded from latents $\boldsymbol{z}$ and corrupted with Gaussian noise, such that $\boldsymbol{o} \sim \mathcal{N}(\boldsymbol{C}\boldsymbol{z} + \boldsymbol{b}, \boldsymbol{\Sigma})$, with $\boldsymbol{C}$ the readout matrix, $\boldsymbol{b}$ a vector of biases, and $\boldsymbol{\Sigma}$ a diagonal matrix of variances. This yields the following log-likelihood function:

$$\log P(\boldsymbol{o}_t|\boldsymbol{z}) = -\frac{1}{2}\left[(\boldsymbol{C}\boldsymbol{z}_t + \boldsymbol{b} - \boldsymbol{o}_t)^T\boldsymbol{\Sigma}^{-1}(\boldsymbol{C}\boldsymbol{z}_t + \boldsymbol{b} - \boldsymbol{o}_t) + n_o\log(2\pi) + \sum_i \log \Sigma_{ii}\right] \tag{S8}$$

The Jacobian of this expression is as follows :

$$\frac{\partial \log P(\boldsymbol{o}_t|\boldsymbol{z}_t)}{\partial \boldsymbol{z}_t} = -\boldsymbol{C}^T\boldsymbol{\Sigma}^{-1}(\boldsymbol{C}\boldsymbol{z}_t + \boldsymbol{b} - \boldsymbol{o}) \tag{S9}$$

Finally, the Hessian is given by :

$$\frac{\partial^2 \log P(\boldsymbol{o}_t|\boldsymbol{z}_t)}{\partial \boldsymbol{z}_t \partial \boldsymbol{z}_t^T} = -\boldsymbol{C}^T\boldsymbol{\Sigma}^{-1}\boldsymbol{C} \tag{S10}$$

## D.2 POISSON LIKELIHOOD

To model spike trains, we assume that they are generated by a Poisson process with an underlying positive rate function for neuron $i$ given by:

$$\mu_i = \beta_i f((\boldsymbol{C}\boldsymbol{z})_i + b_i)\Delta \tag{S11}$$

where $f : \mathbb{R}^n \to \mathbb{R}^n_+$ is a nonlinear function (chosen to be an exponential when modelling the monkey recordings, and a soft ReLU-like nonlinearity elsewhere), $\Delta$ denotes the time bin size, and $\beta_i$ is a neuron-specific gain parameter. This yields the following log-likelihood :

$$\log P(\boldsymbol{o}_t | \boldsymbol{z}_t) = \sum_{i=1}^{n_o} (o_i \log \mu_i - \mu_i + \log o_i!) \tag{S12}$$

where the sum is performed over neurons. Using the shorthand notations $h(x) = \log f(x)$ and $\boldsymbol{a}_t = \boldsymbol{C}\boldsymbol{z}_t + \boldsymbol{b}$, the Jacobian and Hessian of this expression are given by :

$$\frac{\partial \log P(\boldsymbol{o}_t | \boldsymbol{z}_t)}{\partial \boldsymbol{z}_t} = \boldsymbol{C}^T \left[ \boldsymbol{o}_t \odot h'(\boldsymbol{a}_t) - \Delta \boldsymbol{\beta} \odot f'(\boldsymbol{a}_t) \right] \tag{S13}$$

$$\frac{\partial^2 \log P(\boldsymbol{o}_t | \boldsymbol{z}_t)}{\partial \boldsymbol{z}_t \partial \boldsymbol{z}_t^T} = \boldsymbol{C}^T \left[ \text{diag}(\boldsymbol{o}_t \odot h''(\boldsymbol{a}_t)) - \Delta \, \text{diag}(\boldsymbol{\beta} \odot f''(\boldsymbol{a}_t)) \right] \boldsymbol{C} \tag{S14}$$

## E  iLQR ALGORITHM

Our recognition model makes use of the iterative Linear Quadratic Regulator algorithm (iLQR; Li and Todorov, 2004; Tassa et al., 2014) to find the mean of the posterior distribution $q_\phi(\boldsymbol{u}|\boldsymbol{o})$. Iterative LQR is used to solve finite-horizon optimal control problems with non-linear dynamics and non-quadratic costs by (i) linearizing the dynamics locally around some initial trajectory, (ii) performing a quadratic approximation to the control cost around that same trajectory, (iii) solving the linear-quadratic problem generated by the local approximation to obtain better control inputs, and (iv) repeat until convergence, each time linearizing around the trajectory induced by the new inputs. Below, we first introduce the linear-quadratic regulator (LQR), and detail the approximation used in iLQR to turn any non-linear non-quadratic problem into one that can be solved with LQR. Moreover, we provide pseudo-code for our implementation of iLQR (see Algorithm 1).

The Linear Quadratic Regulator is concerned with finding the set of controls $\boldsymbol{u} \in \mathbb{R}^m$ that minimize a quadratic cost function $\mathcal{C}(\boldsymbol{u})$ under deterministic linear dynamics, given by:

$$\mathcal{C}(\boldsymbol{u}) = \sum_{t=0}^{T} \frac{1}{2} \left( \boldsymbol{z}_t^T \boldsymbol{C}_t^{zz} \boldsymbol{z}_t + \boldsymbol{u}_t^T \boldsymbol{C}_t^{uu} \boldsymbol{u}_t + \boldsymbol{z}_t^T \boldsymbol{C}_t^{zu} \boldsymbol{u}_t + \boldsymbol{u}_t^T \boldsymbol{C}_t^{uz} \boldsymbol{z}_t \right) + \boldsymbol{z}_t^T \boldsymbol{c}_t^z + \boldsymbol{u}_t^T \boldsymbol{c}_t^u \tag{S15}$$

$$\text{s.t. } \boldsymbol{z}_{t+1} = \boldsymbol{A}_t \boldsymbol{z}_t + \boldsymbol{B}_t \boldsymbol{u}_t + \boldsymbol{h}_t. \tag{S16}$$

Here, $\boldsymbol{A}_t \in \mathbb{R}^{n \times n}$ is a (possibly time-dependent) transition matrix, $\boldsymbol{B}_t \in \mathbb{R}^{n \times m}$ represents the input channels at time t, and $\boldsymbol{h}_t$ is a state and input-independent term. Note that $\boldsymbol{z} \in \mathbb{R}^n$ is a deterministic function of the initial condition $\boldsymbol{z}_0$ and the sequence of inputs $\boldsymbol{u}$. LQR finds the inputs minimizing Equation S15 using a dynamic programming approach, by recursively finding the feedback rule $(\boldsymbol{K}_t, \boldsymbol{k}_t)$ which gives the optimal inputs to minimize the cost-to-go at each time t as $\boldsymbol{u}_t = \boldsymbol{K}_t \boldsymbol{z}_t + \boldsymbol{k}_t$. Details can be found in `function Backward` in Algorithm 1.

iLQR is an extension of LQR to general dynamics and cost functions. Specifically, iLQR minimizes

$$\mathcal{C}_\theta(\boldsymbol{u}) = \sum_{t=0}^{T-1} r_\theta(\boldsymbol{z}_t, \boldsymbol{u}_t, t) \quad \text{subject to} \quad \boldsymbol{z}_{t+1} = \boldsymbol{f}_\theta(\boldsymbol{z}_t, \boldsymbol{u}_t, t) \tag{S17}$$

where $\theta$ denotes a set of parameters. At iteration $i$, iLQR approximates both the dynamics and the cost around the current trajectory $\boldsymbol{\tau}^i = (\boldsymbol{z}^i, \boldsymbol{u}^i)$ as:

$$\tilde{\boldsymbol{f}}_\theta^i(\delta\boldsymbol{z}_t, \delta\boldsymbol{u}_t, t) \approx \boldsymbol{f}_\theta(\boldsymbol{\tau}^i) + (\nabla_{\boldsymbol{z}} \boldsymbol{f}_\theta)^T \delta\boldsymbol{z}_t + (\nabla_{\boldsymbol{u}} \boldsymbol{f}_\theta)^T \delta\boldsymbol{u}_t \tag{S18}$$

and

$$\tilde{r}_\theta^i(\delta\boldsymbol{z}_t, \delta\boldsymbol{u}_t, t) \approx r_\theta(\boldsymbol{\tau}_t^i) + \frac{1}{2} \left[ \delta\boldsymbol{z}_t^T (\nabla_{zz}^2 r_\theta) \delta\boldsymbol{z}_t + 2\delta\boldsymbol{z}_t^T (\nabla_{zu}^2 r_\theta) \delta\boldsymbol{u}_t + \delta\boldsymbol{u}_t^T (\nabla_{uu}^2 r_\theta) \delta\boldsymbol{u}_t \right] \tag{S19}$$

$$+ \delta\boldsymbol{z}^T (\nabla_z r_\theta) + \delta\boldsymbol{u}^T (\nabla_u r_\theta) \tag{S20}$$

Here, $\delta\boldsymbol{z}$ and $\delta\boldsymbol{u}$ refer to perturbations around the current nominal trajectory, and all $\nabla$ operators correspond to partial differentiation evaluated at the current nominal trajectory $(\boldsymbol{u}^i, \boldsymbol{z}^i)$ and corresponding time $t$.

The above equations are readily identified as a local LQR problem of the form of Equation S15, which can thus be solved using standard dynamic programming tools. Once $\delta \boldsymbol{u}^\star$ minimizing Equation S19 has been computed, the inputs are updated as $\boldsymbol{u}^{i+1} = \boldsymbol{u}^i + \delta \boldsymbol{u}^\star$, and the new state trajectory follows from simulating the dynamics forward with these new inputs. After each LQR update, we thus obtain a new trajectory $\boldsymbol{\tau}^{i+1}$ and the process repeats until convergence to some locally optimal trajectory $\boldsymbol{\tau}^\star$.

Implementation details can be found in Algorithm 1. Note that the backward LQR pass involves inversion of the matrix $\boldsymbol{Q}_{uu}$ (defined in Algorithm 1 `function Backward`). Depending on the specific form of the iLQR cost function, this might not always be positive-definite. Therefore, we include an adaptive Levenberg-Marquard-type regularizer (not described in the pseudo-code) $\boldsymbol{Q}_{uu} \leftarrow \boldsymbol{Q}_{uu} + \lambda \boldsymbol{I}$ to maintain positive definiteness. Thus, iLQR effectively reverts to first-order gradient descent, as opposed to second-order optimization, whenever the locally quadratic approximation is a bad one.

## F    Differentiating through iLQR

Here we discuss how to efficiently differentiate through the iLQR algorithm. This becomes necessary when one wishes to differentiate through a function involving an iLQRsolve, such as the posterior mean of our recognition model (Equation 11). While a naive but simple strategy to achieve this would be to unroll the algorithm and gather gradients for every step, this is expensive both computationally and memory-wise. Amos et al. (2018) derived a way to analytically obtain gradients with respect to the parameters of iLQR, at the cost of a single LQR pass. Specifically, differentiating through an iLQRsolve is achieved by running iLQR to convergence, forming a linear-quadratic approximation around the converged trajectory, following the steps described in Appendix E and differentiating through the corresponding LQR problem. Below, we provide an alternative derivation to Amos et al.'s of the gradients of an LQR solution.

### F.1    LQR optimality conditions

We now introduce use the more compact notation $\boldsymbol{\tau}_t = \begin{bmatrix} \boldsymbol{z}_t \\ \boldsymbol{u}_t \end{bmatrix}, \boldsymbol{F}_t = [\boldsymbol{A}_t \quad \boldsymbol{B}_t], \boldsymbol{C}_t = \begin{bmatrix} \boldsymbol{C}_t^{zz} & \boldsymbol{C}_t^{uz} \\ \boldsymbol{C}_t^{zu} & \boldsymbol{C}_t^{uu} \end{bmatrix}$, which will be used in the rest of this section.

As described in Appendix E, the finite-horizon, discrete-time LQR problem involves minimizing:

$$\mathcal{J} = \sum_{t=0}^{T} \left( \frac{1}{2} \boldsymbol{\tau}_t^T \boldsymbol{C}_t \boldsymbol{\tau}_t + \boldsymbol{c}_t^T \boldsymbol{\tau}_t \right) \tag{S21}$$

subject to constraints on its dynamics

$$\boldsymbol{\tau}_{t+1} = \boldsymbol{F}_t \boldsymbol{\tau}_t + \boldsymbol{f}_t, \tag{S22}$$

following the notation from Appendix E. To solve this problem, we write down the Lagrangian:

$$\mathcal{L} = \sum_{t=0}^{T} \left( \frac{1}{2} \boldsymbol{\tau}_t^T \boldsymbol{C}_t \boldsymbol{\tau}_t + \boldsymbol{c}_t^T \boldsymbol{\tau}_t \right) + \sum_{t=0}^{T-1} \boldsymbol{\lambda}_{t+1}^T (\boldsymbol{F}_t \boldsymbol{\tau}_t + \boldsymbol{f}_t - \boldsymbol{\tau}_{t+1}), \tag{S23}$$

where $\boldsymbol{\lambda}_1, \boldsymbol{\lambda}_2, \cdots, \boldsymbol{\lambda}_T$ are adjoint (dual) variables that enforce the dynamic constraint. Differentiating with respect to $\boldsymbol{\lambda}_t$ and $\boldsymbol{\tau}_t$ enables us to obtain the set of equations satified by $\boldsymbol{\lambda}$ and $\boldsymbol{\tau}$, also known as the KKT conditions (Kuhn and Tucker, 2014; Karush, 2014; Boyd et al., 2004):

$$\begin{bmatrix} \boldsymbol{I} & \boldsymbol{0} \end{bmatrix} (\boldsymbol{C}_t \boldsymbol{\tau}_t + \boldsymbol{c}_t) + \boldsymbol{F}_t^T \boldsymbol{\lambda}_{t+1} - \boldsymbol{\lambda}_t = \boldsymbol{0} \tag{S24}$$

$$\boldsymbol{F}_t \boldsymbol{\tau}_t + \boldsymbol{f}_t - \begin{bmatrix} \boldsymbol{I} & \boldsymbol{0} \end{bmatrix} \boldsymbol{\tau}_{t+1} = \boldsymbol{0} \tag{S25}$$

$$\boldsymbol{C}_T \boldsymbol{\tau}_T + \boldsymbol{c}_T - \begin{bmatrix} \boldsymbol{I} & \boldsymbol{0} \end{bmatrix}^T \boldsymbol{\lambda}_T = \boldsymbol{0} \tag{S26}$$

$$\boldsymbol{z}_0 - \begin{bmatrix} \boldsymbol{I} & \boldsymbol{0} \end{bmatrix} \boldsymbol{\tau}_0 = \boldsymbol{0} \tag{S27}$$

---

**Algorithm 1** iLQRsolve($\mathcal{C}_\theta(\boldsymbol{u}), \boldsymbol{u}^{\text{init}}$)), with $\boldsymbol{u} \in \mathbb{R}^m$ and $\mathcal{C}_\theta$ defined in Equation S17.
**Parameters**: $\theta, \gamma$

---

▶ **iLQR**
  $\boldsymbol{\tau}^0 = \text{Rollout}(\boldsymbol{u}^{\text{init}})$
  **for** $i = 1$ to *converged* **do**
      **for** $t = 0$ to $T$ **do**
          $\boldsymbol{F}_t^z = \nabla_z \boldsymbol{f}_\theta, \boldsymbol{F}_t^u = \nabla_u \boldsymbol{f}_\theta$
          $\boldsymbol{c}_t^z = \nabla_z r_\theta, \boldsymbol{c}_t^u = \nabla_u r_\theta, \boldsymbol{C}_t^{zz} = \nabla_z^2 r_\theta, \boldsymbol{C}_t^{uu} = \nabla_u^2 r_\theta, \boldsymbol{C}_t^{uz} = \nabla_{uz}^2 r_\theta$
      **end for**
      $\boldsymbol{k}_{[0:T-1]}, \boldsymbol{K}_{[0:T-1]} = \text{Backward}(\boldsymbol{F}_{[0:T]}^z, \boldsymbol{F}_{[0:T]}^u, \boldsymbol{c}_{[0:T]}^z, \boldsymbol{c}_{[0:T]}^u, \boldsymbol{C}_{[0:T]}^{zz}, \boldsymbol{C}_{[0:T]}^{uz}, \boldsymbol{C}_{[0:T]}^{uu}))$
      $\boldsymbol{\tau}^i = \text{Forward}(\boldsymbol{K}_{[0:T-1]}, \boldsymbol{k}_{[0:T-1]}, \boldsymbol{\tau}^{i-1})$
  **end for**

◇ **function** Rollout($\boldsymbol{u}$)
  $\boldsymbol{z}_0 = \boldsymbol{0}$
  **for** $t = 1$ to $T$ **do**
      $\boldsymbol{z}_{t+1} = \boldsymbol{f}_\theta(\boldsymbol{z}_t, \boldsymbol{u}_t)$
  **end for**
  **return** $\boldsymbol{\tau} = \{\boldsymbol{z}, \boldsymbol{u}\}$

◇ **function** Backward($\boldsymbol{F}_{[0:T]}^z, \boldsymbol{F}_{[0:T]}^u, \boldsymbol{c}_{[0:T]}^z, \boldsymbol{c}_{[0:T]}^u, \boldsymbol{C}_{[0:T]}^{zz}, \boldsymbol{C}_{[0:T]}^{uz}, \boldsymbol{C}_{[0:T]}^{uu}$)
  $\boldsymbol{v}_T = \boldsymbol{c}_T^z, \boldsymbol{V}_T = \boldsymbol{C}_T^{zz}$
  **for** $t = T - 1$ to $0$ **do**
      $\boldsymbol{Q}_t^{zz} = \boldsymbol{C}_t^{zz} + \boldsymbol{F}_t^{z\top} \boldsymbol{V}_{t+1} \boldsymbol{F}_t^z$
      $\boldsymbol{Q}_t^{uz} = \boldsymbol{C}_t^{uz} + \boldsymbol{F}_t^{u\top} \boldsymbol{V}_{t+1} \boldsymbol{F}_t^z$
      $\boldsymbol{Q}_t^{uu} = \boldsymbol{C}_t^{uu} + \boldsymbol{F}_t^{u\top} \boldsymbol{V}_{t+1} \boldsymbol{F}_t^u$
      $\boldsymbol{q}_t^z = \boldsymbol{c}_t^z + \boldsymbol{F}^z \boldsymbol{v}_{t+1}$
      $\boldsymbol{q}_t^u = \boldsymbol{c}_t^u + \boldsymbol{F}^u \boldsymbol{v}_{t+1}$
      $\boldsymbol{K}_t = -(\boldsymbol{Q}_t^{uu})^{-1} \boldsymbol{Q}_t^{uz}$
      $\boldsymbol{k}_t = -(\boldsymbol{Q}_t^{uu})^{-1} \boldsymbol{q}_t^u$
      $\boldsymbol{V}_t = \boldsymbol{Q}_t^{zz} + \boldsymbol{Q}_t^{zu} \boldsymbol{K}_t$
      $\boldsymbol{v}_t = \boldsymbol{q}_t^z + \boldsymbol{K}_t^T \boldsymbol{q}_t^u$
  **end for**
  **return** $\boldsymbol{k}_{[0:T-1]}, \boldsymbol{K}_{[0:T-1]}$

◇ **function** Forward($\boldsymbol{K}_{[0:T-1]}, \boldsymbol{k}_{[0:T-1]}, \boldsymbol{\tau} = \{\boldsymbol{u}, \boldsymbol{z}\}$)
  $\alpha = 1$
  **repeat**
      $\hat{\boldsymbol{z}}_0 = \boldsymbol{0}$
      $\hat{\boldsymbol{u}}_0 = \alpha \boldsymbol{k}_0$
      **for** $t = 1$ to $T$ **do**
          $\hat{\boldsymbol{z}}_t = f_\theta(\hat{\boldsymbol{z}}_{t-1}, \hat{\boldsymbol{u}}_{t-1})$
          $\hat{\boldsymbol{u}}_t = \boldsymbol{K}_t(\hat{\boldsymbol{z}}_t - \boldsymbol{z}_t) + \alpha \boldsymbol{k}_t$
      **end for**
      $\alpha = \gamma \alpha$
  **until** $\mathcal{C}_\theta(\hat{\boldsymbol{u}}) < \mathcal{C}_\theta(\boldsymbol{u})$
  **return** $\boldsymbol{\tau} = \{\hat{\boldsymbol{z}}, \hat{\boldsymbol{u}}\}$

---

Rearranging, we can rewrite the KKT conditions in matrix form as:

$$
\underbrace{\begin{pmatrix}
\ddots & & & & & \\
& C_t & F_t^T & & & \\
& F_t & 0 & [-I\ 0] & & \\
& & [-I\ 0]^T & C_{t+1} & F_{t+1}^T & \\
& & & F_{t+1} & 0 & \\
& & & & & \ddots
\end{pmatrix}}_{K}
\underbrace{\begin{pmatrix}
\vdots \\
\tau_t \\
\lambda_{t+1} \\
\tau_{t+1} \\
\lambda_{t+2} \\
\vdots
\end{pmatrix}}_{p}
= -
\underbrace{\begin{pmatrix}
\vdots \\
c_t \\
f_t \\
c_{t+1} \\
f_{t+1} \\
\vdots
\end{pmatrix}}_{q}
\tag{S28}
$$

These optimality conditions are satisfied for the solution to the optimization problem $p^\star = (\tau_0^\star, \cdots, \tau_T^\star, \lambda_1^\star, \cdots \lambda_T^\star)$. Equation S28 implies that the solution of the LQR problem $p^\star$ will satisfy:

$$
p^\star = -K^{-1}q. \tag{S29}
$$

Computing this quickly becomes infeasible as $K$ grows with long-time horizons, and Equation S28 is typically solved in linear time using a dynamic programming approach, as described in Appendix E.

## F.2 Backpropagating through the LQR solver

Differentiating through an LQR solve boils down to differentiating through the backsolve in Equation S29. In the following, we denote the adjoint of parameter $\theta$ as $\bar{\theta}$. From Giles (2008), we know that the adjoint of the backsolve operation is given by:

$$
\bar{q} = -K^{-T}\bar{p}, \tag{S30}
$$
$$
\bar{K} = -K^{-T}\bar{p}p^T = \bar{q}p^T. \tag{S31}
$$

We note that Equation S30 has the same form as Equation S29, which means we can compute $\bar{q} = (\cdots, \bar{c}_t, \bar{f}_t, \cdots)^T$ by solving another LQR problem. After solving for $\bar{\tau}$, we can then compute $\bar{K}$ as an outer-product of $\bar{z}$ with $y$ to get:

$$
\underbrace{\begin{pmatrix}
\ddots & & & & & \\
& \bar{K}_{C_t} & \bar{K}_{F_t^T} & & & \\
& \bar{K}_{F_t} & & & & \\
& & & \bar{K}_{C_{t+1}} & \bar{K}_{F_{t+1}^T} & \\
& & & \bar{K}_{F_{t+1}} & & \\
& & & & & \ddots
\end{pmatrix}}_{\bar{K}}
\tag{S32}
$$

$$
= \begin{pmatrix}
\vdots \\
\bar{c}_t \\
\bar{f}_t \\
\bar{c}_{t+1} \\
\bar{f}_{t+1} \\
\vdots
\end{pmatrix}
\begin{pmatrix}
\cdots & \tau_t^{\star T} & \lambda_{t+1}^{\star}{}^T & \tau_{t+1}^{\star}{}^T & \lambda_{t+2}^{\star}{}^T & \cdots
\end{pmatrix}
\tag{S33}
$$

$$
= \begin{pmatrix}
\ddots & & & & & \\
& \bar{c}_t\tau_t^{\star T} & \bar{c}_t\lambda_t^{\star T} & & & \\
& \bar{f}_{t+1}\tau_t^{\star T} & & & & \\
& & & \bar{c}_{t+1}\tau_{t+1}^{\star}{}^T & \bar{c}_{t+1}\lambda_{t+1}^{\star}{}^T & \\
& & & \bar{c}_{t+1}\lambda_{t+1}^{\star}{}^T & & \\
& & & & & \ddots
\end{pmatrix}.
\tag{S34}
$$

Collecting all the gradients of $\bar{C}_t$ and $\bar{F}_t$, we arrive at

$$\bar{C}_t = \frac{1}{2}(\bar{K}_{C_t} + \bar{K}_{C_t}^T) = \frac{1}{2}(\bar{c}_t \tau_t^{\star T} + \tau_t^\star \bar{c}_t^T) \tag{S35}$$

$$\bar{F}_t = \bar{K}_{F_t} + \bar{K}_{F_t^T}^T = \bar{f}_{t+1} \tau_t^{\star T} + \lambda_{t+1}^\star \bar{c}_t^T. \tag{S36}$$

Note that we have symmetrized the adjoint of $C_t$, which ensures that $C_t$ remains symmetric after each gradient update. The antisymmetric part of $C_t$ does not contribute to the LQR cost.

Finally, one subtlety arises from the fact that Equation S22 and Equation S21 are written as a function of $\tau$ in the general LQR setting. In the iLQR case however, the LQR problem is local at each iteration, and $\delta\tau$ vanishes at convergence. If we denote by i the last iteration before declaring convergence, one can however write the problem as a function of the variable of interest $\tau^\star$, using :

$$\mathcal{J} = \sum_{t=0}^{T} \left( \frac{1}{2}(\tau_t^{\star T} - \tau_t^{iT}) C_t (\tau_t^\star - \tau_t^i) + c_t^T (\tau_t^\star - \tau_t^i) \right) \tag{S37}$$

$$= \sum_{t=0}^{T} \left( \frac{1}{2}\tau_t^{\star T} C_t \tau_t^\star + (c_t^T - \tau_t^i C_t) \tau_t^\star \right) + \text{cst} \tag{S38}$$

subject to constraints on its dynamics

$$\tau_{t+1}^\star = F_t \tau_t^\star + f_t - F_t \tau_t^i. \tag{S39}$$

This implies that the values for $C$ and $c$ need to be adjusted accordingly, such as to reflect the switch of variable from $\delta\tau$ during the optimization to the fixed point $\tau^\star$ to compute gradients. Note that at convergence we can use $\tau^i \approx \tau^\star$, giving access to all the necessary variables to compute gradients with respect to $\theta$.

# G   DETAILS OF EXPERIMENT 1

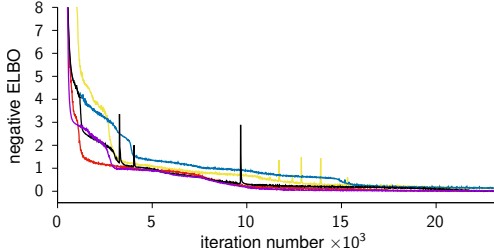

Figure S2: Illustration of the learning curves for 5 different runs of the "forced autonomous" iLQR-VAE model. The model consistently gets stuck in plateaus during the optimization, leading to a slower convergence than their "free" counterparts (see Figure 1).

The data in Section 3.1 was generated from an autonomous linear dynamical system with $n = 8$, $m = 3$, and $n_o = 8$ where $n_o$ is the dimension of the observation space. All the models were fit using the dynamics within the ground-truth model class, i.e with linear dynamics, $n = 8$, $m = 3$, and $n_o = 8$. We optimized the model parameters with Adam, using (manually optimized) learning rates of $0.04/(1+ \sqrt{k/1})$ for the free iLQR-VAE model, $0.04/(1+ \sqrt{k/1}$ for autonomous iLQR-VAE and $0.02/(1+ \sqrt{k/30}$ for LFADS, where k is the iteration number. We used GRU networks with 32 units to parametrize the LFADS encoders (one encoder for the initial condition and one for the inputs). Note that while all methods run in similar wallclock time in this example, this will ultimately be implementation and data-dependent.

In Figure S2, we show additional learning curves for the "forced autonomous" models; these show that, even for different initializations and trajectories through the loss landscape, the model consistently gets stuck in plateaus. This can be contrasted with the free-form iLQR-VAE models.

# H   FURTHER DETAILS OF LORENZ ATTRACTOR

The chaotic Lorenz attractor consists of a three-dimensional state $(\ell_1, \ell_2, \ell_3)$ evolving according to

$$\dot{\ell}_1 = 10(\ell_2 - \ell_1) \qquad \dot{\ell}_2 = \ell_1(28 - \ell_3) - \ell_2 \qquad \dot{\ell}_3 = \ell_1\ell_2 - 8\ell_3/3 \tag{S40}$$

For our example, we generated data by integrating Equation S40 over a long time period using a Runge-Kutta solver (RK4) followed by z-scoring and splitting the resulting state trajectory into 112 non-overlapping bouts (Figure 2A). We added Gaussian noise with a standard deviation of 0.1, and trained iLQR-VAE on this dataset (Figure 2B, bottom). We then fitted these data using GRU dynamics with $n = 20$ and $m = 5$.

The normalized $k$-step mean-squared error was defined as follows:

$$\text{MSE}_k = \sum_{t=0}^{T-k} \|\boldsymbol{x}_{t+k} - \hat{\boldsymbol{x}}_{t+k}\|^2 \tag{S41}$$

$$R_k^2 = 1 - \frac{\text{MSE}_k}{\sum_{t=0}^{T-k} \|\boldsymbol{x}_{t+k} - \bar{\boldsymbol{x}}\|^2} \tag{S42}$$

where $\hat{\boldsymbol{x}}_{t+k}$ is the prediction at time $t + k$, and $\bar{\boldsymbol{x}}$ the mean for this trial.

## I    LEARNING INPUT-DRIVEN NONLINEAR DYNAMICS

To bridge the gap between autonomous nonlinear dynamics (see Section 3.2) and real data, we evaluated iLQR-VAE on an *input-driven* nonlinear system, the Duffing oscillator. We generated Duffing trajectories that included a perturbation of the Duffing state half-way through. We then embedded those into the spiking activity of 200 neurons (see below). We found that iLQR-VAE could successfully learn the dynamics and infer the timing of the perturbations.

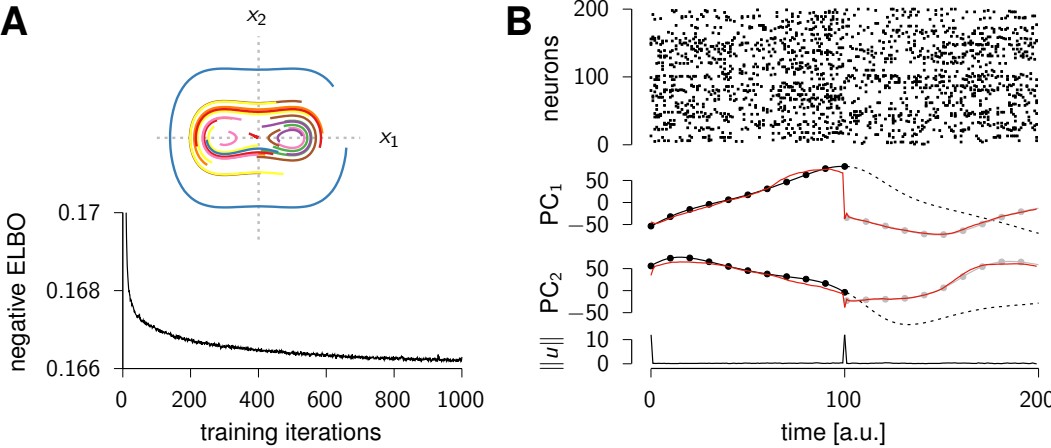

Figure S3: **(A)** Top: example Duffing trajectories (100 time steps, $dt = 0.03$) starting from random initial conditions. Bottom: negative ELBO during the course of training. **(B)** Top: spike raster corresponding to an example test sample. Middle: first two principal components of the posterior mean over firing rates (red), given the spiking data shown at the top. For comparison, the ground-truth PCs in the first half of the trial (before the perturbation) are shown as black dots, with their hypothetical unperturbed continuation shown as a dashed line. The second half of the ground-truth PCs (after the perturbation) are shown in gray. Bottom: norm of the inferred input.

The dynamics of the Duffing oscillator are given by

$$\dot{x}_1 = x_2 \qquad \dot{x}_2 = x_1 - x_1^3. \tag{S43}$$

To generate each training sample, we integrated Equation S43 from *two* different random initial conditions for 100 time steps each using a Runge-Kutta solver (RK4) with $dt = 0.03$. Example such trajectories are shown in Figure S3A (top); note that each trajectory can be understood as the evolution of the system in state-space for a given energy level of the oscillator. We then concatenated those two trajectories to yield a single trajectory of 200 steps with a perturbation in the middle. We then linearly mapped the low-dimensional oscillator onto a 200-dimensional state, before passing it through the nonlinearity of Equation S7 to obtain a set of firing rates, which then gave rise to observations via a Poisson process (Figure S3B, top).

We generated 112 training and 112 testing trials in this way. We fit these data using iLQR-VAE with $n = 20$, $m = 4$, and found that it could successfully infer the latent trajectories (see Figure S3B, middle). Importantly, iLQR-VAE learned to fit most of the trajectories as an autonomously evolving dynamical system, and only used inputs to explain the sudden change in the oscillator's energy level triggered by the perturbation (see Figure S3B, bottom). This shows that the model can successfully disentangle ongoing dynamics from external inputs, suggesting that it is well-suited for identifying input-driven dynamics in real data.

## J    COMPARISON OF LFADS AND iLQR-VAE ON A TOY INPUT INFERENCE TASK

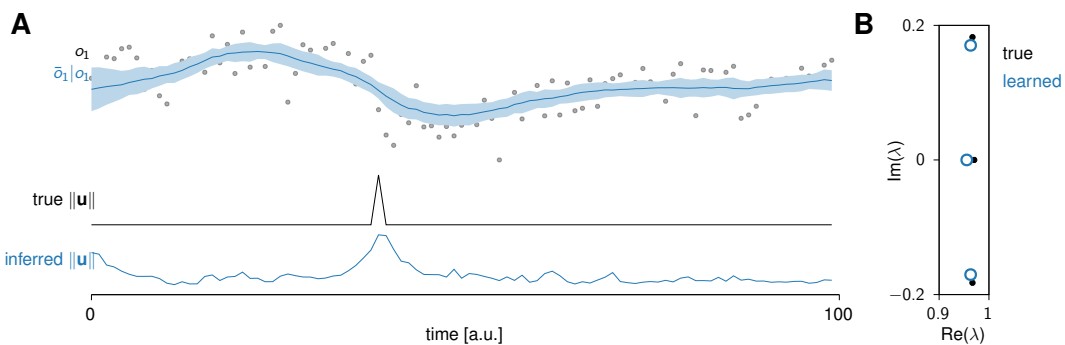

Figure S4:  **Details of the sparse input inference in LFADS and iLQR-VAE. (A)** Top: example observations (black dots) and inferred posterior mean (blue line). Bottom: true and inferred inputs. LFADS can infer the timing of the largest input, but also uses non-zero inputs during the rest of the trial. **(B)** Comparison of the true (black) and learned (blue) eigenvalue spectra.

We used the LFADS implementation from `https://github.com/google-research/computation-thru-dynamics/tree/master/lfads_tutorial`, which we modified to include linear dynamics and Gaussian likelihoods. We then evaluated the quality of the input reconstruction by measuring how much input variance was captured by the models. We report this as the $R^2$ from inferred to true inputs.

We used a generative model within the ground truth model class. For each dataset, we performed a hyper-parameter search to choose the best-performing encoder architecture and learning rate for LFADS.

Results of this experiment are summarized in Table S1. iLQR-VAE – which did not require any hyperparameter tuning for these examples – inferred inputs more accurately for all dataset sizes and trial lengths.

|  | LFADS | iLQR-VAE |
|---|---|---|
| S 1x1000 | $0.05 \pm 0.02$ | $0.94 \pm 0.01$ |
| S 10x100 | $0.15 \pm 0.06$ | $0.84 \pm 0.01$ |
| S 32x100 | $0.29 \pm 0.01$ | $0.83 \pm 0.05$ |
| S 56x100 | $0.31 \pm 0.08$ | $0.80 \pm 0.02$ |
| S 10x200 | $0.27 \pm 0.01$ | $0.93 \pm 0.01$ |
| AR 56x100 | $0.28 \pm 0.02$ | $0.81 \pm 0.02$ |

Table S1: Comparison of iLQR-VAE and LFADS on 6 input inference tasks. Results are reported as $R^2$ (mean $\pm$ sem) over 3 random seeds for each.

Our results suggest that LFADS' performance improves with larger amounts of data. More surprisingly, LFADS also seems to perform better when the data is split into shorter trials. In particular, we found it difficult to fit LFADS on the single long trial, but the dynamics could be learned more accurately if this data was split into 10 trials of 100 steps. On the other hand, iLQR-VAE inferred

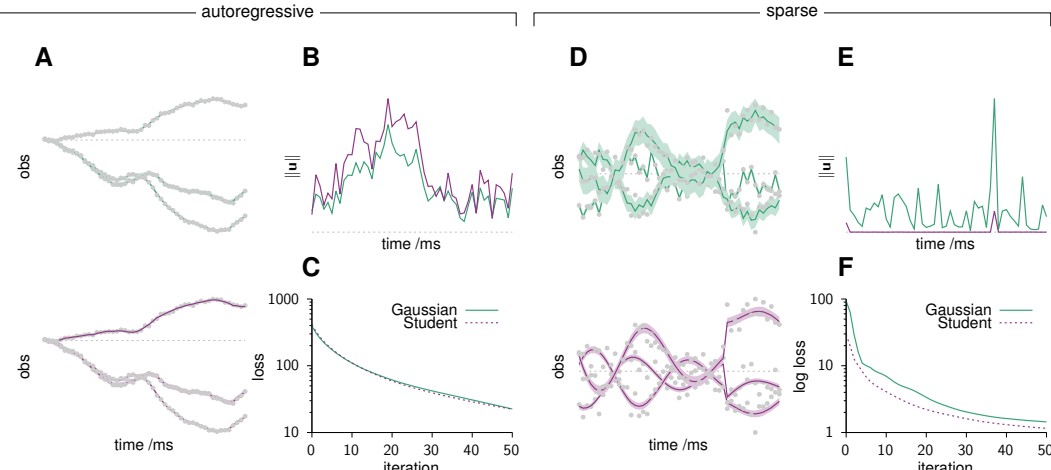

Figure S5: Comparison of the Student (pink) and Gaussian (green) prior for learning a linear dynamical system driven by autoregressive inputs (AR) or sparse inputs. **(A)** Fit of the observations using the Gaussian (top) and Student (bottom) priors. Both allow to fit the observations highly accurately. **(B)** Inferred input norm for both priors. The temporal structure of the signal is very similar in both cases, as the Student prior essentially becomes Gaussian for large values of $\nu$. **(C)** Evolution of the loss for both choices of prior. The loss curves are closely aligned, and both models converge to a similar ELBO value. **(D)** Fit of the observations using the Gaussian (top) and Student (bottom) priors. Both models allow to fit the observations, but the Student prior allows to obtain smoother trajectories. **(E)** Inferred input norm for both priors. The Student prior is close to 0 at all times, except when it requires a sharp input to explain the data. On the other hand, the Gaussian prior requires a large variance to be able to fit sparse inputs, leading to non-zero inferred inputs at all time points. **(F)** Evolution of the loss for both prior choices. The loss curve for the Student prior is lower than the Gaussian one throughout training, and converges to a higher ELBO value.

inputs more accurately for longer trials. This is what we would expect if the model is well learnt, as longer trials contain more information to fit the inputs accurately.

One important distinction between the two methods, which partly explains LFADS' lower $R^2$, is the prior it over inputs (auto-regressive prior for LFADS and Student for iLQR-VAE). In Figure S4 we show an example of LFADS, on one of the test examples of the S 56x100 dataset. In this example, LFADS infers its largest input concurrently to the ground truth input, but also infers small inputs when there are none in the ground truth. This has a significant impact on the $R^2$ metric. Note however that this is not the only effect at play here, as emphasized by the lower performance on the AR dataset. The impact of the choice of prior in iLQR-VAE is discussed further in Appendix K.

## K    COMPARISON OF THE STUDENT AND GAUSSIAN PRIORS

In this section, we compare the performance of the Gaussian and Student priors on two toy examples. The first consists of data generated by a linear dynamical system ($n = 3$, $m = 3$, $n_o = 10$, Gaussian likelihood) driven by autoregressive Gaussian inputs (close to the Gaussian prior). The second one uses the same system, but driven by sparse inputs (closer to the Student prior). We find that in the first example, both priors yield extremely similar results (see Figure S5(A-C)). Indeed, the Student prior learns a very high value of $\nu \sim 20$, thus becoming nearly Gaussian.

In the sparse input case however, the Student prior allows to fit the data considerably better. As we can see in Figure S5(D-F), the Gaussian prior learns a large variance to fit the sparse inputs, leading to higher baseline noise than in the true system.

As shown here, the Student prior offers a more flexible model, as the Gaussian case is recovered for large $\nu$ values. Note however that using the Gaussian prior ensures that the input term in the iLQR cost function is always convex in u, which can facilitate the optimization and allow iLQR to

converge faster in some cases. Moreover, in the case of autonomous dynamics (e.g Lorenz attractor and Maze dataset) both priors will converge to the same solution.

## L   FURTHER DETAILS OF SINGLE TRIAL ANALYSES

### L.1   BENCHMARKING AGAINST EXISTING METHODS

To allow for direct comparison with benchmarks reported Pei et al. (2021), we first used data provided by the Neural Latents Benchmark (NLB) challenge, available at https://gui.dandiarchive.org/#/dandiset/000128.

We used 1720 training trials and 510 validation trials, which were drawn randomly for each instantiation of the model to avoid overfitting to test data. The risk of overfitting to the dataset was lowered by the fact that iLQR-VAE requires very little hyperparameter optimization. For this experiment, we fitted iLQR-VAE to the neural activity using a model with MGU dynamics ($n = 60$), a Student prior over inputs ($m = 15$), and a Poisson likelihood ($n_o = 182$ neurons). We trained models on trials spanning all reach conditions and restricting data to a time window starting 250 ms before and ending 450 ms after movement onset, binned at 5ms. For regression to hand velocity, we introduced a lag of 100ms between neural activity and hand velocity. As the test data used in the NLB challenge is not publicly available, the results we reported were not computed on the exact same data split. However, the model performed highly consistently across random seeds, such that we expect iLQR-VAE's performance to be directly comparable to the results from Pei et al. (2021). To fit these data, we ran iLQR-VAE on 168 CPUs for $\sim$ 6h, using a mini-batch size of 168 trials.

The co-smoothing metric used to assess how well the model fit the data is defined as log-likelihood score :

$$\frac{1}{n_{sp} \log 2} \left( \mathcal{L}(\boldsymbol{\lambda}; \hat{\boldsymbol{y}}_{n,t}) - \mathcal{L}(\hat{\boldsymbol{\lambda}}_n, \hat{\boldsymbol{y}}_{n,t}) \right) \tag{S44}$$

where the overall log-likelihood $\mathcal{L}$ is the sum of all the log-likelihoods evaluated at all points and for all neurons, $\boldsymbol{\lambda}$ denotes the vector inferred time-varying firing rates, $\hat{\boldsymbol{\lambda}}_n$ is the mean firing rate for neuron $n$ and $n_{sp}$ is the total number of spikes.

### L.2   FURTHER ANALYSES

A key feature of monkey M1 motor cortical recordings is the prevalence of rotational dynamics in the data (Churchland et al., 2012). These can be captured using jPCA, a method developed to find the subspace in which the dynamics are most rotational, which was recently generalized by Rutten et al. (2020). Here, we found that we could uncover clean rotational dynamics from the single-trial firing rates, similarly to Pandarinath et al. (2018).

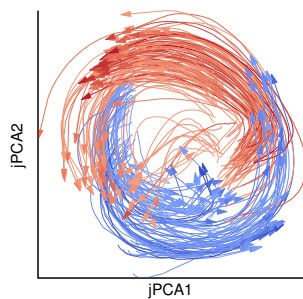

Figure S6: Projection of the neural activity of 200 movements in the subspace defined by the top two jPCA axes. jPCA finds the subspace capturing most rotations in the data, while spanning the same space as the top 2 principal components. Here, the jPCA subspace was found using the single-trial firing rates. Projection of the neural activity yields very clean rotational trajectories.

## M   FURTHER DETAILS OF THE CONTINUOUS REACHING TASK ANALYSIS

### M.1   DETAILS OF THE ANALYSES

For our analyses of the primate data in Section 3.4, we considered the first 22 minutes of the recording session 'indy_20160426' from O'Doherty et al. (2018). We binned spikes at 25 ms resolution

and considered all neurons with a firing rate of at least 2Hz. Behavioural data took the form of the velocity of the hand of the monkey in the xy-plane and were extracted as the first derivative of a cubic spline fitted to the position over time. We z-scored the hand velocity and shifted it by 120ms, following on Jensen et al. (2021).

To fit iLQR-VAE, the resulting dataset was divided into 336 non-overlapping pseudo-trials of which a random half were used to fit the generative model and the other half of the trials were used as a held-out test dataset. We fitted a model with $n = 50, m = 10$ using the non-linear dynamics described in Equation S4. The latent state was then mapped onto both the kinematics and neural observations. We used a linear readout from latents to 2D kinematics variables, and a linear readout following by a nonlinearity from latents to the firing rates of 130 neurons.

After fitting the iLQR-VAE to neural activity and behavior jointly, we then proceeded to infer $u$ from neural activity alone. Next, we computed the kinematic reconstruction error on the test dataset as the fraction of variance captured in both x- and y- hand velocities.

Finally, we analyzed the inputs to the model after fitting, in relation to specific events in the task and the behaviour. We defined 'movement onset' after each target onset as the time at which the hand speed first exceeded $0.03 \, \text{m s}^{-1}$. We aligned the z-scored input $u$ on each trial to target onset and movement onset separately for visualization purposes. We performed a similar analysis for hand speed and mean z-scored neural activity which were also z-scored and aligned to target and movement onset for comparison with the control input.

## M.2   COMPARISON OF iLQR-VAE AND bGPFA

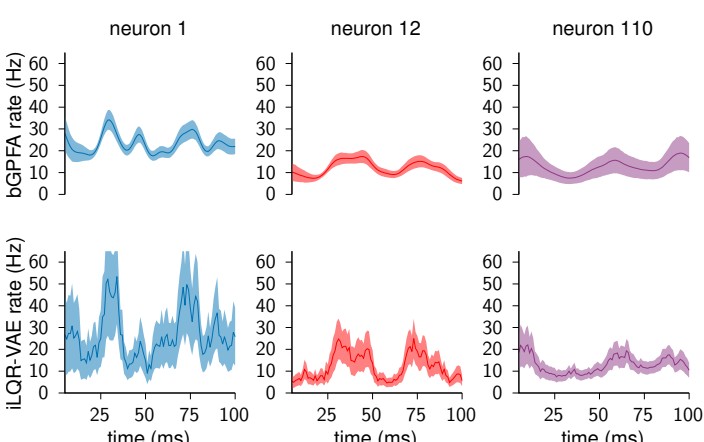

Figure S7: Comparison of the firing rates inferred by bGPFA and iLQR-VAE on the first 100ms of the continuous reaching task data, for three different neurons. bGPFA learns smoother trajectories. On the other hand, fitting iLQR-VAE with a Gaussian prior with no temporal structure allows to capture more variance in the firing rates, which in turn leads to a better decoding of the kinematics.

As a further way of understanding the relative benefits and disadvantages of iLQR-VAE, we compared its performance with bGPFA, a fully Bayesian extension of GPFA (Yu et al., 2009) that enables the use of non-Gaussian likelihoods, scales to very large datasets, and was recently shown to outperform standard GPFA on this same continuous reaching dataset (Jensen et al., 2021). Importantly, bGPFA makes different assumptions to iLQR-VAE, as it places a smooth prior directly on the latents with no explicit notion of dynamics. We fit both methods using 10 minutes of data (chunked into pseudo-trials for iLQR-VAE and as a continuous trial for bGPFA). For iLQR-VAE we then performed inference and retrained the posterior covariance on the first minute of data whilst fixing the generative parameters. We found that while both methods captured similar trends in the firing rates, bGPFA yielded smoother estimates, but iLQR-VAE captured larger modulations (consistent with the higher $R^2$ when regressing from firing rates to hand velocity). Note that the firing rate estimates here are not as smooth as for the Maze dataset (c.f. Figure 4A), because iLQR-VAE was fit using a Gaussian prior over inputs with non-zero variance at all times, effectively implying an autoregressive prior on the latent trajectories and firing rates.

From Figure S7, one can notice that bGPFA struggles to capture larger variations in the firing rate. This suggests oversmoothing, and might explain why the method does not capture hand kinematics

as well as iLQR-VAE ($R^2 = 0.6$ for bGPFA and $0.76$ for iLQR-VAE.) This is indeed what we see in Figure S8.

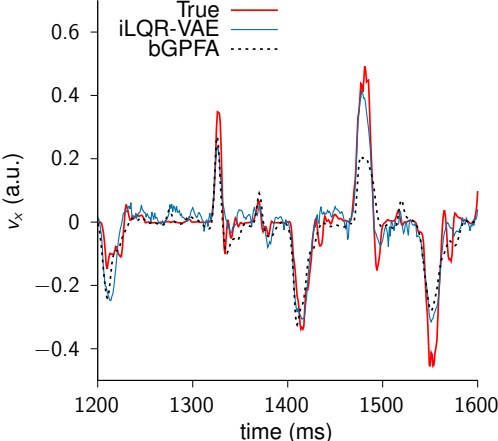

Figure S8: Ground truth hand velocity (red) and decoded kinematics using iLQR-VAE (blue) and bGPFA (dotted black). A linear decoder was trained on 9 minutes of data and then evaluated on the first minute of the recording (shown here). bGPFA struggles to capture the biggest peaks in the velocity, consistent with the smoother firing rates and the lower $R^2$.

## N    LINK TO KALMAN FILTERING

The Linear Quadratic Regulator and the Kalman filter (Kalman, 1964) are algorithms designed for systems with linear dynamics and Gaussian noise. LQR finds the optimal feedback control law to minimize a cost $\mathcal{C}$ in deterministic systems, while the Kalman filter yields an estimate of the state from observations corrupted with process and observation noise. It is well-known that Kalman filtering and LQR are dual of one another, and they can both be combined into LQG to yield an optimal control law from noisy observations. Here, we explore another link between LQR and Kalman smoothing, by showing how LQR can be used as a Kalman smoother. Moreover, in order to gain insights into the learning process of iLQR-VAE, we explore different procedures for learning the parameters of a Kalman filter.

LINEAR QUADRATIC CONTROL AS FILTERING

The Kalman smoother assumes dynamics of the form

$$z_{t+1} = Az_t + B^K w_t \tag{S45}$$
$$o_t = Cz_t + v_t \tag{S46}$$

with $w \sim \mathcal{N}(0, I)$, $v \sim \mathcal{N}(0, \Sigma_v)$, and the initial condition is assumed to be generated by a Gaussian distribution with known parameters $z_1 \sim \mathcal{N}(\mu, \Pi)$.

On the other hand, LQR assumes the following fully-deterministic dynamics :

$$z_{t+1} = Az_t + B^I u_t \tag{S47}$$
$$o_t = Cz_t \tag{S48}$$

with $z_1$ known exactly.

Note than in the iLQR-VAE framework we have thus far only considered cases where no observed external inputs were given. However these can be straightforwardly included as an additional $\hat{B}\hat{u}$ term in Equation S47 and Equation S45.

The Kalman smoother's objective is to minimize the expected mean squared error between the inferred latent state and the true state, $\mathbb{E}\left[\|x - \hat{x}\|^2\right]$. As described in Aravkin et al. (2017), with linear dynamics and Gaussian noise, this becomes equivalent to minimizing the following objective w.r.t $z$:

$$\mathcal{L}(z) = \|\Pi^{-1/2}(z_1 - \mu)\|^2 + \sum_{t=1}^{T-1} \|B^K(z_{t+1} - Az_t)\|^2 + \sum_{t=1}^{T} \|\Sigma_v^{-1/2}(o_t - Cz_t)\|^2 \tag{S49}$$

where the first two terms correspond to the prior over the initial condition and smoothness of the trajectory, and the last term represents the likelihood of the observations. Interestingly, this can be related to the objective we minimize to find the posterior mean in iLQR-VAE (Equation 11):

$$\mathcal{L}(\boldsymbol{u}) = \|\boldsymbol{\Sigma}_0^{-1/2}\boldsymbol{u}_0\|^2 + \sum_{t=1}^{T-1}\|\boldsymbol{\Sigma}_u^{-1/2}\boldsymbol{u}_t\|^2 + \sum_{t=1}^{T}\|\boldsymbol{\Sigma}_{\boldsymbol{v}}^{-1/2}(\boldsymbol{o}_t - \boldsymbol{C}\boldsymbol{z}_t)\|^2 \tag{S50}$$

$$= \|\boldsymbol{\Sigma}_0^{-1/2}\boldsymbol{u}_0\|^2 + \sum_{t=1}^{T-1}\|\boldsymbol{\Sigma}_u^{-1/2}(\boldsymbol{z}_{t+1} - \boldsymbol{A}\boldsymbol{z}_t)\|^2 + \sum_{t=1}^{T}\|\boldsymbol{\Sigma}_{\boldsymbol{v}}^{-1/2}(\boldsymbol{o}_t - \boldsymbol{C}\boldsymbol{z}_t)\|^2. \tag{S51}$$

The right-hand sides of Equation S49 and Equation S50 become identical when $\boldsymbol{\Sigma}_u = \boldsymbol{B}^K$ and $\boldsymbol{\Sigma}_0 = \boldsymbol{\Pi}$. Note that the introduction of the $\boldsymbol{B}^I$ matrix in Equation S45 unties the two formulations slightly by allowing for further mixing between the input channels that isn't accounted for by the prior. In the examples we consider next, we therefore set $\boldsymbol{B}^I = \boldsymbol{I}$.

The above equations show how LQR can be used to solve the standard Kalman filtering problem, with the key difference being that the optimization is performed over inputs $\boldsymbol{u} = \{\boldsymbol{u}_0, \dots, \boldsymbol{u}_{T-1}\}$ rather than latent trajectories $\boldsymbol{z} = \{\boldsymbol{z}_1, \dots, \boldsymbol{z}_T\}$ directly. This is illustrated in Figure S9(A), where a Rauch-Kung-Striebel (RKS) smoother and LQR were ran on the same set of 8-dimensional observations arising from an 8-dimensional linear dynamical system, and inferred the same latent trajectory given the ground-truth parameters. As we only use LQR to parametrize the mean of the posterior distribution, we trained the recognition model for 100 steps to get the uncertainty over the latents, which was very similar to the output of the RKS smoother.

LEARNING A KALMAN FILTER

We then proceeded to learn the parameters of the models using either iLQR-VAE, an Expectation-Maximization (EM) procedure, or direct minimization of the negative log likelihood of the data (Figure S9B-C).

Interestingly, the EM algorithm is closely related to iLQR-VAE, since the E-step finds the latent trajectories minimizing Equation S49, when iLQR-VAE solves Equation S50 in an inner optimization loop. While there exists an analytical solution for the M-step in the case of the Kalman filter, this does not generalize to nonlinear dynamics and non-Gaussian noise. Therefore, we used a gradient descent procedure for the maximization step.

Both of these were performed using Adam with a learning rate of 0.02, and with initial parameters drawn from the same distributions. We see in Figure S9C that iLQR-VAE reaches a smaller NLL in considerably fewer iterations than gradient descent, which we hypothesize is due to the good preconditioning given by iLQR (discussed in Figure 1). Note however that the cost of one iLQR-VAE iteration is higher than the direct computation of Equation S49.

In this section, we have shown in a simple linear-quadratic example how iLQR-VAE performs filtering by inferring the process noise as inputs. While this is undoubtedly an unconventional approach, it becomes particularly valuable in cases where dynamics are non-linear and the noise non-Gaussian. Indeed, in such cases the problem of learning an estimator for the latent state is a very difficult one, typically solved using methods such as particle filtering or unscented Kalman filters (Doucet and Johansen, 2009; Wan et al., 2001). iLQR-VAE offers another way to solve this problem, with close links to the aforementioned approaches.

## O    ANALYSIS OF THE INFERENCE GAP

In order to evaluate the benefits of defining the recognition model implicitly through the generative parameters, we compared iLQR-VAE to a more standard sequential variational auto-encoder, using a bidirectional recurrent neural network as the recognition model. We generated data from the same system as in Figure 3, in the form of 76 trials of 100 time steps. We used the same generative model in both cases (linear dynamics with $n = 3$, $m = 3$, $n_o = 10$, Student prior), such that the only difference lay in the choice of recognition model. We compared the ELBO to a more accurate estimate of the log-likelihood, the Importance Weighted Autoencoder (IWAE) bound (Burda et al.,

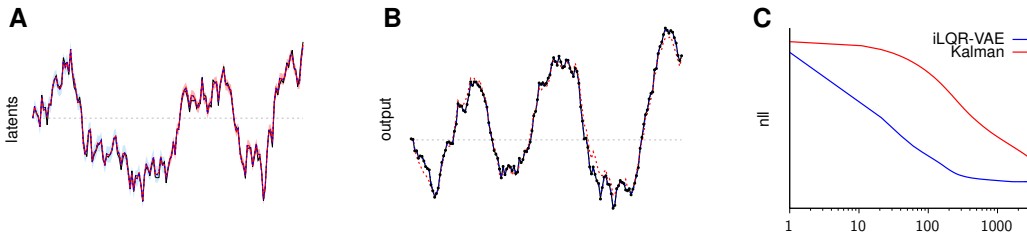

Figure S9: **Comparison of the posterior mean of a Kalman smoother and LQR.** We ran both LQR and a Kalman smoother on noisy observations generated by a latent system (see text for details). **(A)** The Kalman smoother (blue) and LQR (dotted red) both inferred the same posterior mean for the latent trajectories, matching the true latents (black) almost perfectly. The posterior uncertainty is shown for both cases on half of the data. The iLQR-VAE uncertainty was obtained by optimizing the variance of the recognition model for 200 iterations, and then drawing 1000 samples from the recognition model. **(B)** We compared learning the parameters of the posterior distribution using iLQR v.s direct minimization of the NLL. On unseen data, both were able to converge to a solution close to the smoothed output trajectory. Note that we stopped the optimization after 8000 iterations. **(C)** Learning curves of the direct optimization and iLQR-VAE on the same 168 training trials. Note that we used Adam with the same learning rate of 0.02 in both cases, in order to directly compare the effect of the gradient steps. Both x and y axes use a logarithmic scale.

2015), which is computed as

$$\mathcal{L}_{\text{IWAE}} = \mathbb{E}_{u_1,..u_n \sim q(u|o)} \left[ \log \left( \frac{1}{k} \sum_{i=1}^{k} \frac{p(o, z_i)}{q(u_i|o)} \right) \right] \tag{S52}$$

where we used Monte-Carlo sampling with 1000 samples to evaluate the expectation. This then allowed us to compute the *inference gap* (Cremer et al., 2018) of both models as $\mathcal{L}_{IWAE} - ELBO$. As shown in Figure S10, iLQR-VAE has a smaller inference gap throughout training, leading to faster and more robust convergence. This confirms the intuition that keeping the recognition and generative models in sync throughout training reduces the inference gap.

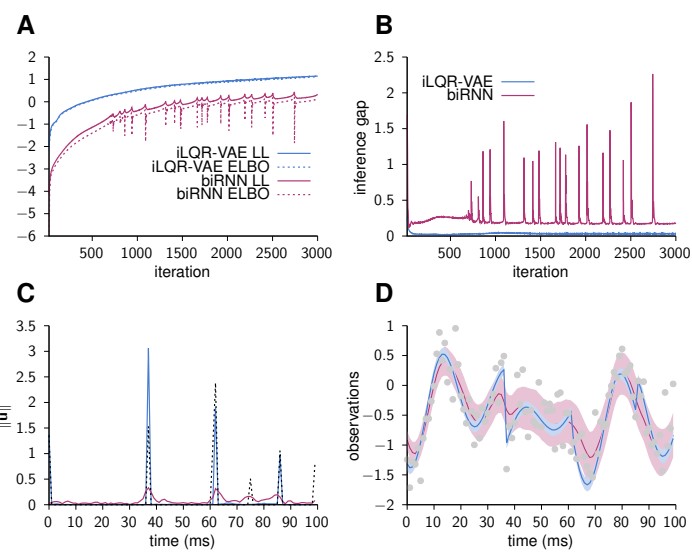

Figure S10: **Comparison of iLQR with a biRNN recognition model.** **(A-B)** Loss and inference gap as a function of iteration number (starting from iteration 20) for iLQR-VAE (blue) and the biRNN model (pink). **(C)** Inferred input norm at the end of training for iLQR-VAE and the biRNN model. Ground truth input is shown in dotted black lines. **(D)** Inferred output at the end of training for iLQR-VAE and the biRNN model. Both models explain the observations (grey dots) well, but iLQR-VAE captures the sharp transitions better.

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
