# OpenReview forum: "iLQR-VAE : control-based learning of input-driven dynamics with applications to neural data"
_ICLR.cc/2022/Conference — ICLR 2022 Oral_

### Official Review · Reviewer_YcP3 · 2021-10-26

**Correctness:** 4
**Technical Novelty And Significance:** 3
**Empirical Novelty And Significance:** 2
**Recommendation:** 8
**Confidence:** 3

**Main Review:**

strengths
---------
The problem of simultaneously learning dynamics and ongoing external inputs is a crucial one, especially in fields such as neuroscience where complex dynamical systems are often analyzed through incomplete measurements of the system (i.e. not every neuron is monitored). This paper presents a sound approach to tackling this problem, though I admit my understanding if the control field is limited.

The paper is clearly written and easy to follow.

I appreciate building up the complexity of the toy examples.

Comaprison to state-of-the-art methods is nice and clear.


weaknesses
----------
Fig. 1: "Indeed, iLQR is unable to find initial conditions that would explain the data well, resulting in a much higher initial loss." While the paper implicitly points to this outcome as an advantage of the more flexible prior, it is concerning that iLQR cannot find the initial condition of this very simple example. Is this an issue with the iLQR algorithm itself? The chosen prior? It would be very useful to understand this behavior in the simple example before moving on to more complex examples.

The leap from example 1 (nonlinear autonomous dynamics) and example 2 (linear dynamics + sparse inputs) to the real data (presumably nonlinear dynamics + ongoing inputs) begs the question: how well does the model perform with (nonlinear dynamics + sparse inputs)? Showing good performance in this regime would increase my confidence in the model's ability to handle more complex, real-world data.

There is very little reference to related work, especially with regards to inferring control inputs in dynamical systems (linear or nonlinear). The paper would feel much more complete with this information (which could be provided as an additional appendix if space is short).

The general problem of decoupling ongoing dynamics from external inputs is underdetermined, and results rely heavily on the structure imposed on these aspects of the model. In this paper the prior on the inputs is a sparse one, but what happens if there is mismatch between this prior and the real data? How will that affect the results (and their interpretation)? The paper comments on this several times (and shows how an AR prior in LFADS performs when inputs are sparse, but not the reverse), but I'd appreciate a more thorough discussion of this issue in the Discussion.

Learning in the proposed model seems to have some strong parallels with the EM algorithm, except that maxima from iLQR replace expectations. Is this true? Could be interesting to briefly draw this connection when describing the inference strategy.


minor
-----
Fig. 1 caption: first sentence, add "system" to end?

paragraph that starts "At the beginning...", second sentence, typo: "learning consists in _making_"

same paragraph: "We note that this regime is facilitated here by our choice of generator dynamics, which we initialised to be very weak initially and therefore easily controllable." What does it mean for the generator dynamics to be "initialised to be very weak"? This is potentially an interesting point that is unclear to me as written. What does "weak initialisation" mean in this context?

**Summary Of The Paper:**

This paper presents a new approach for inference in a model that simultaneously provides latent dynamics, initial conditions, and - importantly - external inputs. This approach is enabled by using the outcome of an optimization algorithm (iLQR) in the recognition model, recently enabled by other work in the field. The paper demonstrates the use of this approach on simulated data and real neural data, and compares to other classic and contemporary models of nonlinear dynamical systems.

**Summary Of The Review:**

I'd lean towards accepting this paper, due to its novelty and thoroughness. I'd be willing to argue even more strongly for it if the authors could address some of the points above (especially the more complex toy example).

---

### Official Review · Reviewer_gF5H · 2021-11-01

**Correctness:** 4
**Technical Novelty And Significance:** 4
**Empirical Novelty And Significance:** 4
**Recommendation:** 8
**Confidence:** 4

**Main Review:**

Post-rebuttal: Thanks for addressing my concerns. The authors have updated the manuscript and added experiments. I will not change my recommendation.

##########

Pros:

+ Implicit recognition model implied by the generative model. It incorporates the inferred forward dynamics.

+ Competitive performance to STOA method

+ Small parameter space for easy hyperparameter tuning

Concerns:

- The method is extended and compared to LFADS. How about with other methods?

- Unlike the bidirectional RNN in LFADS, the implicit recognition model should incorporate the dynamics from the trained generative model. This bias could be good or bad for the training. Can the authors address more?

- There are other methods like VIND consider to incorporate the dynamics from the trained generative model. This better be compared or discussed.

- Is this method fast or slow comparing to LFADS or other methods?

- How well can the proposed method learn other typical types of dynamical systems such as fixed attractors, continuous attractors and etc.

Minor:

* Typo: "LFDADS" in paragrah 2, page 2

**Summary Of The Paper:**

The paper proposes a control-based variational inference approach that learns latent neural dynamics in input-driven SSM. It utilizes iLQR in the recognition model that transforms it into an optimal-control problem. The recognition model in the proposed method is implicitly implied by the generative model and thus reduces the number of free parameters comparing to existing methods. The proposed methods are evaluated on sythetic chaotic attractor and real-world neural recordings.


**Summary Of The Review:**

Overall, I vote for accepting. The paper is well written. I like the idea of transforming to control problem. The implicit recognition model also interesting. This would be useful to learn neural dynamics from large population recordings. Hopefully the authors can address my concern in the rebuttal period.

---

### Official Review · Reviewer_Ve7m · 2021-11-01

**Correctness:** 4
**Technical Novelty And Significance:** 4
**Empirical Novelty And Significance:** 3
**Recommendation:** 8
**Confidence:** 4

**Main Review:**

Authors present a novel method for simultaneously learning latent dynamics and inferring unobserved control inputs. The method performs very well on several toy datasets (autonomous and input-driven linear dynamical system + Lorenz attractor), is on par with state of the art methods on benchmark datasets for neural data analysis methods, and allows for better reconstruction of hand kinematics for primate recordings in a continuous reaching task. The main novelty of the method lies in the utilization of iLQR with implicit differentiation. The paper also extensively discusses LFADS and provides very interesting insights on LFADS.

An important component  of the method is the prior on input distribution. Authors show how to use a Gaussian prior or a multivariate Student prior (it allows for strong inputs when needed, represents the fact that inputs come as shared events and are spatially and temporally independent, and authors mention it might be one advantage over LFADS's autoregressive prior). This is a key idea of the paper and I would like to see more experiments highlighting the importance of the prior. I would recommend that the authors show (possibly in the appendix), the differences between the inferred inputs and performances when using different priors.

The derivation of the ELBO does not seem straightforward to me. To go from equation 7 to 8, don't you need to condition p(o|u) on z_0 as well? Maybe I am not correct, but I would appreciate if authors clarify this point (and correct the manuscript if needed)

About the computational complexity. The model is linear in T and cubic in n, which seems fast for low-dimensional latents only. Authors mention that iQLR-VAE enables fast learning of dynamics, but they also mention it as a limitation in the discussion. Could authors clarify this? Moreover, I believe the computational complexity comparison with LFADS in Figure 1 bottom left is not fair. Authors compare the number of training iteration without comparing the complexity of each iteration. I would like to see comparison for other datasets as well.

There are many performance metrics used in this paper to show that iQLR-VAE performs as well as other state of the art methods. However, there are no comparisons of the processes learnt by the different methods. Showing superimposed processes inferred by different methods  (not only LFADS) for several experiments would allow to gain insight into the methods and how they compare (in term of uncertainty, smoothness, etc...). Similarly, for Figure 4, it would be interesting to see a low dimensional representation of neural activity (and comparisons of different methods). For this experiment, is it necessary to use a latent processes of dimension 50? Wouldn't it be sufficient and more efficient to use lower dimensional latent for this task?

Minor point :
 - In figure 4 caption, there is a typo and the last part should be panel D not C.


**Summary Of The Paper:**

This paper proposes ILQR-VAE, a novel method that allows to simultaneously learn latent dynamics and infer unobserved control inputs. The method relies on IQLR solver and recent advances allowing for implicit differentiation to maximize an Evidence Lower Bound on log-likelihood of observation to infer a conditional distribution over inputs as well as latent states.
Authors show comparisons to other models (the closest one being LFADS) on toy datasets and benchmark datasets for neural data analysis methods.
iLQR-VAE is on par with state-of-the-art methods on many datasets, does not require any extensive hyperparameter optimization, and allows for fast inference when dimension of the latent processes is not too high.

**Summary Of The Review:**

I recommend acceptance of the paper. This is a very interesting and novel method, that is shown to perform very well on several different datasets.
There are many quantitative experiments but showing qualitative comparisons would be very beneficial for gaining insights into how different methods compare.

---

### Decision · Program_Chairs · 2022-01-20

**Decision:**

Accept (Oral)

**Comment:**

The paper introduces a novel control-based variational inference approach that learns latent dynamics in an *input-driven* state-space model. An optimal control solution (iLQR) is implicitly used as the recognition model which is fast and compact. Reviewers unanimously agree on the high quality writing and high significance of the work. This paper advances the horizon of nonlinear dynamical system models with unobserved input, an impactful contribution to the neuroscience and time series communities.